# Temporal dynamics and developmental memory of 3D chromatin architecture at *Hox* gene loci

Daan Noordermeer[1], Marion Leleu[1], Patrick Schorderet[1,2], Elisabeth Joye[1], Fabienne Chabaud[3], Denis Duboule[1,3]*

[1]School of Life Sciences, Ecole Polytechnique Fédérale de Lausanne, Lausanne, Switzerland; [2]Department of Molecular Biology, Harvard University, Boston, United States; [3]Department of Genetics and Evolution, University of Geneva, Geneva, Switzerland

**Abstract** *Hox* genes are essential regulators of embryonic development. Their step-wise transcriptional activation follows their genomic topology and the various states of activation are subsequently memorized into domains of progressively overlapping gene products. We have analyzed the 3D chromatin organization of *Hox* clusters during their early activation in vivo, using high-resolution circular chromosome conformation capture. Initially, *Hox* clusters are organized as single chromatin compartments containing all genes and bivalent chromatin marks. Transcriptional activation is associated with a dynamic bi-modal 3D organization, whereby the genes switch autonomously from an inactive to an active compartment. These local 3D dynamics occur within a framework of constitutive interactions within the surrounding Topological Associated Domains, indicating that this regulation process is mostly cluster intrinsic. The step-wise progression in time is fixed at various body levels and thus can account for the chromatin architectures previously described at a later stage for different anterior to posterior levels.

*For correspondence: denis.duboule@epfl.ch

**Competing interests:** The authors declare that no competing interests exist.

**Reviewing editor**: Robb Krumlauf, Stowers Institute for Medical Research, United States

## Introduction

Mammalian *Hox* genes encode proteins that are essential for patterning along the rostral-to-caudal body axis of the developing embryo (*Duboule and Morata, 1994*; *Krumlauf, 1994*). Mouse and human *Hox* genes are organized in four genomic clusters (*HoxA* to *HoxD*), where the relative position of the genes strongly impacts upon their patterns of expression. This structure-function relationship was initially described in *Drosophila* (*Lewis, 1978*) and further extended to vertebrates (*Gaunt et al., 1988*; *Duboule and Dolle, 1989*; *Graham et al., 1989*), where an additional correspondence exists between gene position and the timing of transcriptional activation ('temporal colinearity', *Izpisua-Belmonte et al., 1991*; *Deschamps and van Nes, 2005*).

In murine embryos, transcription of *Hox* genes can be divided in several phases and is first detected at around embryonic day 7 (E7) at the most posterior aspect of the primitive streak region (*Deschamps and Wijgerde, 1993*; *Forlani et al., 2003*). Over time, *Hox* genes are sequentially activated following their chromosomal order and transcripts encoded by the last *Hox* group 13 genes can be detected at around E9, that is two days after the onset of activation (*Deschamps et al., 1999*; *Kmita and Duboule, 2003*; *Deschamps and van Nes, 2005*). This transcriptional progression (the '*Hox* clock', *Duboule, 1994*) thus extends over several days. In the pre-somitic mesoderm (PSM), this sequential activation needs to be coordinated with the time-sequenced production of body segments (the 'segmentation clock', *Pourquie, 2003*), such that newly produced somites acquire distinct genetic identifiers (*Dubrulle et al., 2001*; *Zakany et al., 2001*). Next, the various states of *Hox* gene activity are fine-tuned and

**eLife digest** Most animals are symmetrical about an imaginary line that runs from the head to the tail. A family of genes called the *Hox* family ensures that the cells in an animal embryo develop into the correct body parts along this head-to-tail axis. *Hox* genes—which are found in animals as diverse as flies and humans—are often clustered on the chromosomes, and their order within a cluster affects when and where each *Hox* gene is 'switched on'.

In mammals, *Hox* genes at one end of a cluster are switched on first and along almost the entire length of the embryo. *Hox* genes near the other end of the cluster are expressed later and only towards the hind end of the animal. And *Hox* genes at the furthest end of the cluster are expressed last and in the very tip of the developing tail. The time when a *Hox* gene is expressed depends largely on its relative position within the gene cluster. However, it is not clear how the ordering of the genes within a cluster is translated into a schedule whereby the genes are sequentially switched on during development.

Much of the DNA in a chromosome is wrapped around proteins to form a structure called chromatin; chromatin is normally tightly packed, but 'unpacking' it allows the genes to be accessed and switched on. Now, Noordermeer et al. have used a technique called 'circular chromosome conformation capture' to follow how the packing of the chromosomes that carry the *Hox* gene clusters changes during embryonic development. Harvesting cells from mouse embryos of different ages, and cross-linking the DNA to the proteins, allowed those genes that are packed in the chromatin to be distinguished from those that have been unpacked and activated.

When the embryo is still just a ball of almost identical cells, all the *Hox* genes are switched off and packed into inactive chromatin. However, Noordermeer et al. found that, as the embryo develops and when each *Hox* gene is switched on in turn, the relevant region of DNA is also unpacked and moved into more active chromatin. This mechanism likely prevents *Hox* genes that direct the development of the hind end of the mouse from being switched on too early, and hence it avoids body parts being misidentified and developing incorrectly. Further, the patterns of active chromatin vs inactive chromatin can be fixed at each section along head-to-tail axis, such that it will be memorized in all daughter cells produced subsequently from each particular body section.

Future challenges will be to uncover the trigger behind the step-wise transition of every *Hox* gene from inactive chromatin to active chromatin, and to crack the underlying 'clock' that controls the timing of this process.

memorized, ultimately leading to domains along the rostral to caudal axis where partially overlapping sets of HOX products can be observed ('spatial colinearity'). As a result, genes located at 3'positions (e.g., groups 3, 4) are transcribed almost along the entire embryonic axis, including the lateral plate mesoderm, paraxial mesoderm and neural tube, whereas the 5'-located group 10 or 11 are active in the posterior trunk and group 13 in the tip of the tail bud only (*Deschamps et al., 1999*; *Kmita and Duboule, 2003*; *Deschamps and van Nes, 2005*).

While both temporal and spatial colinear processes likely reflect one and the same organizational principle, they are nevertheless implemented with distinctive features. Spatial colinearity could be recapitulated by several single-gene transgenes (e.g., [*Puschel et al., 1991*; *Whiting et al., 1991*]), yet not in all instances (*Tschopp et al., 2012*). Indeed, a systematic analysis of modified *HoxD* clusters *in vivo* revealed that, at a late stage, the sustained transcription of these genes at the correct body level primarily relies upon local regulatory elements (*Tschopp et al., 2009*), which are present in transgenic constructs. In contrast, the precise timing of *Hoxd* gene activation depends on the integrity of the full cluster, a genomic situation observed thus far in all animals developing following a temporal rostral to caudal progressive strategy during their early development (*Duboule, 1994*). The genomic clustering of *Hox* genes is thus considered as an essential feature for temporal colinearity to properly process, whereas it may not be as important for the correct distribution of HOX products along the AP-axis, at least in the late phase of spatial colinearity (*Duboule, 2007*; *Tschopp et al., 2009*; *Noordermeer and Duboule, 2013*).

Even though the mechanisms underlying temporal and spatial colinearities are becoming increasingly understood, many aspects of how genomic topology is translated into sequential transcriptional

activation remain to be clarified. In vertebrates, two conceptual frameworks have been proposed to account for this process, the first relying on bio-molecular mechanisms (e.g., *Duboule, 1994*) and the second involving biophysical forces (*Papageorgiou, 2001*). In embryonic stem (ES) cells, that is cells that reflect best the state of *Hox* genes before their activation, *Hox* clusters are decorated by both repressive (H3K27me3) and activating (H3K4me3) marks (*Bernstein et al., 2006*; *Schuettengruber et al., 2007*; *Soshnikova and Duboule, 2009*; *Noordermeer and Duboule, 2013*). Subsequently, cells that activate these genes in a time sequence resolve this bivalent chromatin state and show two opposing distributions of histone marks over the *HoxD* cluster: transcribed genes carry large domains of H3K4me3 marks, whereas inactive genes are covered by H3K27me3 marks only (*Soshnikova and Duboule, 2009*).

The same dichotomy in chromatin marks over *Hox* gene clusters was observed in various parts of the E10.5 embryonic trunk, in parallel with the spatial colinear distribution of transcripts (*Noordermeer et al., 2011*). The analysis of the 3D chromatin organization at this stage revealed a bi-modal compartmentalization, whereby active genes labeled by H3K4me3 are clustered together and physically separated from the inactive genes, labeled by H3K27me3 that are also found in a defined spatial structure (*Noordermeer et al., 2011*). These 3D compartments, whose sizes correlate with the number of active vs inactive genes, may reinforce the proper maintenance of long-term transcriptional states at various AP levels by isolating *Hox* clusters from their surrounding chromatin and reducing interference between the active and inactive chromatin domains. Such distinct bimodal 3D organizations, associated with transcriptional regulation at *Hox* clusters, have been observed in other instances, either in the embryo (*Montavon et al., 2011*; *Andrey et al., 2013*) or in mouse and human cultured cells (*Fraser et al., 2009*; *Ferraiuolo et al., 2010*; *Wang et al., 2011*; *Rousseau et al., 2014*).

However, *in-embryo* conformation studies were reported so far only in the context of spatial colinearity, that is by comparing samples from different body levels at the same developmental stage. Consequently, a potential association between these bimodal chromatin structures and the progressive activation of transcription along the *Hox* gene clusters, rather than its maintenance, remained to be assessed. In this study, we describe the 3D organization of *Hox* gene clusters at high resolution during the implementation of temporal colinearity in the PSM and show that their stepwise activation occurs in parallel with their physical transition from a negative to a positive domain. We also show that this process is accompanied by series of long-range contacts with the flanking gene deserts, even though these contacts remain largely invariable throughout temporal colinearity, unlike what was observed during limb development (*Andrey et al., 2013*). We discuss whether this stepwise transition of genes from one domain to the other may guide temporal colinearity or, in contrast, is a mere consequence of a sequential transcriptional activation.

## Results

### Inactive *Hox* genes in ES cells are organized into a single 3D compartment

In order to monitor the 3D organization of *Hox* clusters during their sequential activation, we considered ES cells as a starting point of our time curve. These cells indeed represent early embryonic cells related to blastocyst inner cell mass cells, that is when *Hox* genes are all supposedly silent. We hypothesized that these cells reflect the ground state 3D architecture of the *Hox* clusters, which we assessed by using high-resolution 4C-seq (Circular Chromosome Conformation Capture; *Noordermeer et al., 2011*; *van de Werken et al., 2012*) and a variety of viewpoints within all four *Hox* clusters. These various baits generated similar interaction profiles with the majority of sequence reads covering the gene clusters and extending within several kilobases (kb) on either sides, as illustrated by *Hoxd13*, *Hoxd9* and *Hoxd4* (*Figure 1A*, *Figure 1—figure supplements 1–4*). Additional contacts were scored in the flanking gene deserts, though with significantly lower frequencies (see section 'Temporal colinearity within a constitutive framework of long-range interactions'). The overall size of the strong interaction profiles exactly matched the distribution of bivalent chromatin marks in these cells, with a moderate level of H3K27me3 covering the cluster and rather weak H3K4me3 peaks labeling promoters (*Figure 1A*, *Figure 1—figure supplements 1–4*; *Bernstein et al., 2006*; *Soshnikova and Duboule, 2009*). Therefore, prior to their activation, *Hox* clusters are already organized into 3D chromatin compartments that physically separate the chromatin decorated by bivalent marks from the genomic surroundings, even though some contacts are established at a

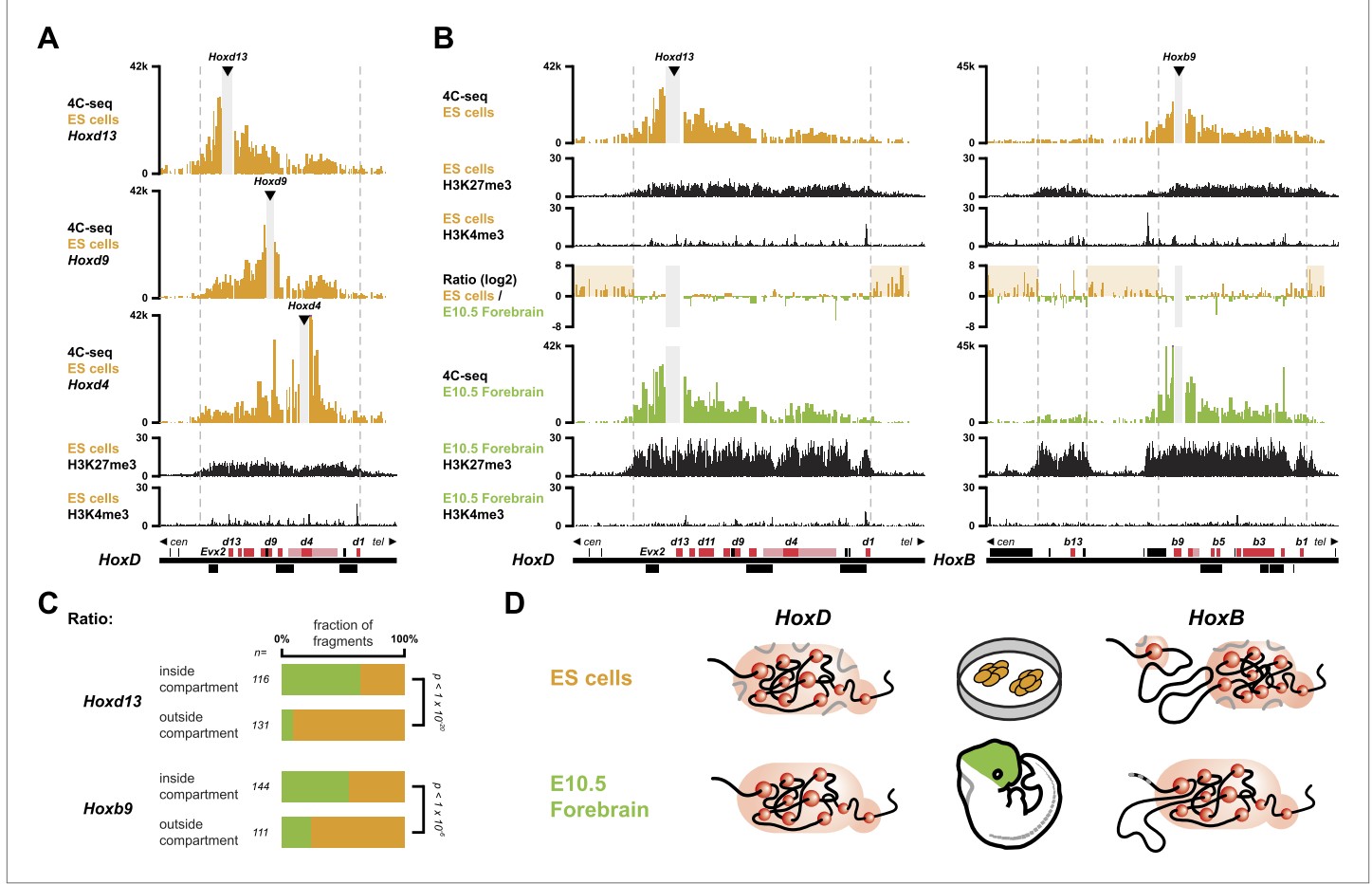

**Figure 1**. *Hox* clusters in ES cells are organized as 3D compartments. (**A**) Quantitative local 4C-seq signal for the *Hoxd13* (top), *Hoxd9* (middle) and *Hoxd4* (bottom) viewpoints in ES cells. Below, the H3K27me3 and H3K4me3 ChIP-seq signals are aligned. The boundaries of the inactive *Hox* gene compartments are indicated by dashed lines. The locations of *Hox* genes (red) and of other transcripts (black) are shown below. (**B**) Quantitative local 4C-seq signal for the *Hoxd13* (left) and *Hoxb9* (right) viewpoints, either in ES (orange) or in E10.5 forebrain (green) cells. Below, the H3K27me3 and H3K4me3 ChIP-seq signals are aligned. Ratios between the 4C-seq signals in ES cells and E10.5 forebrain are indicated between the profiles, with signal in one color indicating that the viewpoint interacts more with this fragment in the sample represented by this color. Regions of increased interactions outside the 3D *Hox* gene compartments in ES cells are highlighted in orange. (**C**) Distribution of ratios inside and outside the inactive 3D *Hox* gene compartments in both ES and E10.5 forebrain cells. Fragments are classified either as positive in ES cells (orange), or positive in E10.5 forebrain cells (green). The number of fragments is indicated below. Significance between distribution inside and outside 3D compartments was calculated using a G-test of independence. (**D**) Model of 3D compartmentalization of the inactive *HoxD* and *HoxB* clusters in both ES cells and E10.5 forebrain cells. The increased contacts with the surrounding chromatin in ES cells are illustrated by invading grey lines.

The following figure supplements are available for figure 1:

**Figure supplement 1**. 3D compartments in the *HoxD* cluster are less discrete in ES cells than in embryonic brain cells.

**Figure supplement 2**. 3D compartments in the *HoxD* and *HoxB* cluster are less discrete in ES cells than in embryonic brain cells.

**Figure supplement 3**. 3D compartments in the *HoxB* cluster are less discrete in ES cells than in embryonic brain cells.

**Figure supplement 4**. 3D compartments in the *HoxC* and *HoxA* cluster are less discrete in ES cells than in embryonic brain cells.

**Figure supplement 5**. Distribution of ratios inside and outside the inactive 3D *Hox* gene compartments in both ES and E10.5 forebrain cells.

**Figure supplement 6**. Different discretion of 3D compartments is not due to overall increased background signal.

**Figure supplement 7**. Increased *Hox* background transcription in ES cells.

larger scale, outside the gene cluster itself (see section 'Temporal colinearity within a constitutive framework of long-range interactions').

These global 3D domains including the *Hox* clusters and their immediate flanking DNAs resemble the chromatin architecture found in embryonic forebrain cells, where the silent *Hox* clusters are covered by high levels of H3K27me3 only (*Noordermeer et al., 2011*). A more quantitative comparison in 3D architectures between ES cells and E10.5 forebrain cells nevertheless indicated that in ES cells, *Hox* genes interacted more with the outside chromatin, relative to their interactions within the cluster, as compared to forebrain cells (*Figure 1B,C*, *Figure 1—figure supplements 1–5*). Therefore, despite the fact that the clusters are presumably inactive in both situations, the presence of bivalent marks in ES cells coincided with a 3D domain that has elevated relative levels of interactions with the directly surrounding regions, when compared to its counterpart in brain cells (*Figure 1D*, left and *Figure 1—figure supplement 6A*). This difference is more pronounced at the *HoxB* cluster (*Figure 1B*, right). In embryonic forebrain cells, *HoxB* forms a single 3D compartment, excluding the 80 kb large repeat-rich intergenic region located between *Hoxb13* and *Hoxb9*, which loops out (*Noordermeer et al., 2011*). In contrast, both the *Hoxb13* and *Hoxb9* viewpoints revealed local 3D compartments in ES cells, matching again the extent of bivalent histone marks, yet these two compartments remained separated and did not fuse. Rather, they displayed increased interactions with the nearby chromatin, as if decreased internal interactions would increase contacts outside the cluster (*Figure 1B,C*, *Figure 1—figure supplements 2, 3 and 5*). In ES cells, the *HoxB* cluster is thus organized in two 3D compartments, which have more interactions with their genomic surroundings than in forebrain cells (*Figure 1D*, right and *Figure 1—figure supplement 6B*).

To have a possibly more unbiased view on how 3D compartments and the presence of H3K27me3 and H3K4me3 modifications relate to each other, in both ES and brain cells, we devised an approach to correlate 4C-seq signals with either H3K27me3 or H3K4me3 ChIP-seq signal (*Table 1*; 'Materials and methods'). In both cell types, H3K27me3 marks strongly correlated with the 3D organization, suggesting a direct link between these two readouts. A considerably lower correlation was scored for *HoxB*, perhaps related to the absence of clustering of the two H3K27me3 marked sub-domains. In contrast, no particular correlation was observed between the 3D organization and the presence of H3K4me3 marks, in the bivalent state (*Table 1*), suggesting that H3K4me3 marks and/or the associated factors do not noticeably contribute to the formation of 3D compartments in ES cells.

### *Hox* genes are transcribed at low levels in ES cells

It was previously reported that genes covered by bivalent marks in ES cells can be transcribed at low levels, resulting in detectable spliced transcripts (*Stock et al., 2007*). We assessed whether the observed difference in the strength and homogeneity of the interaction profiles between ES cells and embryonic brain cells was associated with distinct levels of background transcription. In ES cells, RNA-seq detected transcription for most *Hox* genes, though generally at very low level (*Figure 1—figure supplement 7A,B*). RT-qPCR of a subset of transcripts confirmed that some of these low-level transcripts (particularly the *Hoxd13* and *Hoxb13* transcripts) constitute genuine processed transcripts (*Figure 1—figure supplement 7C*). In contrast, transcription of *Hox* genes in E10.5 forebrain cells was rarely detected, and no reliable spliced transcripts were detected (*Figure 1—figure supplement 7*). Therefore, when *Hox* genes are decorated by bivalent chromatin marks, they appear more permissive for background transcription as compared to other cell types where they are covered by H3K27me3 marks only, likely illustrating the increased resistance to transcription of the latter condition. In this context, posterior *Hox* genes seems to be more prone to background transcriptional activation in ES cells than more anterior *Hox* genes, in contrast to their subsequent dynamics of activation in future embryonic tissues where anterior genes come first. This may reflect the presence of strong enhancers in their vicinity (*Montavon et al., 2011*).

### Dynamics of 3D compartments during sequential *Hox* gene activation

Next, we assessed whether this large 3D domain observed in ES cells is modified when *Hox* genes become activated in the pre-somitic mesoderm (PSM) or instead, whether the previously observed positive and negative compartments are only established at a later stage to fix and memorize particular combinations of *Hox* gene activities determined at earlier stages and at various body levels. For this purpose, we compared the 4C-seq profiles from ES cells with those obtained from early embryonic E8.5 PSM cells dissected out at Theiler stage 13, posterior from the approximate level of the 12[th] to

**Table 1.** Spearman's rank correlation coefficient between pairs of 4C-seq and ChIP-seq samples

| 4C-seq | ChIP-seq | | |
| --- | --- | --- | --- |
| | Input | H3K27me3 | H3K4me3 |
| *Hoxd13* ES cells 1 | −0.14 | **0.52** | 0.24 |
| *Hoxd13* ES cells 2 | −0.07 | **0.40** | 0.22 |
| *Hoxd13* E8.5 PSM | −0.03 | **0.58** | 0.13 |
| *Hoxd13* E10.5 Forebrain 1 | −0.12 | **0.67** | 0.26 |
| *Hoxd13* E10.5 Forebrain 2 | −0.09 | **0.69** | 0.25 |
| *Hoxd13* E10.5 Anterior trunk | −0.07 | **0.80** | 0.30 |
| *Hoxd9* ES cells 1 | −0.08 | **0.63** | 0.28 |
| *Hoxd9* ES cells 2 | −0.13 | **0.59** | 0.26 |
| *Hoxd9* E8.5 PSM | −0.05 | **0.31** | 0.29 |
| *Hoxd9* E10.5 Forebrain 1 | −0.08 | **0.66** | 0.26 |
| *Hoxd9* E10.5 Forebrain 2 | −0.12 | **0.61** | 0.28 |
| *Hoxd9* E10.5 Anterior trunk | −0.15 | **0.67** | 0.47 |
| *Hoxd4* ES cells 1 | 0.01 | **0.48** | 0.11 |
| *Hoxd4* ES cells 2 | −0.07 | **0.50** | 0.29 |
| *Hoxd4* E8.5 PSM | −0.04 | 0.04 | **0.38** |
| *Hoxd4* E10.5 Forebrain 1 | −0.05 | **0.59** | 0.24 |
| *Hoxd4* E10.5 Forebrain 2 | −0.04 | **0.58** | 0.27 |
| *Hoxd4* E10.5 Anterior trunk | −0.07 | 0.16 | **0.59** |
| *Hoxc13* ES cells 1 | −0.03 | **0.39** | 0.20 |
| *Hoxc13* E8.5 PSM | −0.03 | **0.55** | −0.03 |
| *Hoxc13* E10.5 Forebrain 1 | −0.07 | **0.57** | 0.18 |
| *Hoxc13* E10.5 Anterior trunk | −0.05 | **0.82** | 0.00 |
| *Hoxb13* ES cells 1 | −0.05 | **0.12** | 0.02 |
| *Hoxb13* ES cells 2 | −0.08 | −0.01 | **0.15** |
| *Hoxb13* E8.5 PSM | 0.10 | **0.29** | −0.17 |
| *Hoxb13* E10.5 Forebrain 1 | 0.02 | **0.48** | 0.09 |
| *Hoxb13* E10.5 Forebrain 2 | 0.08 | **0.44** | 0.10 |
| *Hoxb13* E10.5 Anterior trunk | −0.03 | **0.49** | 0.26 |
| *Hoxb9* ES cells 1 | 0.01 | **0.47** | 0.09 |
| *Hoxb9* ES cells 2 | 0.03 | **0.34** | 0.04 |
| *Hoxb9* E8.5 PSM | −0.04 | −0.30 | **0.57** |
| *Hoxb9* E10.5 Forebrain 1 | 0.02 | **0.63** | 0.19 |

*Table 1. Continued on next page*

14th forming somite (*Figure 2A*, scheme; *Figure 2—figure supplements 1 and 2*). In the most caudal aspect of this latter cellular territory, transcriptional activation had progressed up to the *Hoxd9* gene, whereas the *Hoxd10* to *Hoxd13* loci remained silent (*Soshnikova and Duboule, 2009*). This cellular population was thus composed of a mixture of cells positive and negative for *Hoxd9* expression, whereas all cells were negative for *Hoxd13*. Conversely, the majority of cells expressed *Hoxd4*. The inactive *Hoxd13* viewpoint interacted mostly with the domain labeled by H3K27me3, at the centromeric side of the cluster (*Figure 2A*, bottom left). In contrast, the active *Hoxd4* gene essentially interacted with the other transcribed genes on the telomeric side of the cluster, labeled by H3K4me3 marks (*Figure 2A*, bottom right). The same bi-modal 3D organization was observed for all *Hox* gene clusters (*Figure 2—figure supplements 1 and 2*).

We correlated 4C-seq signals with ChIP-seq data in both early and late embryonic samples (*Figure 2B*; *Table 1*) and *Hoxd13* always strongly correlated with H3K27me3 histone marks. In contrast, in both the E8.5 tail bud and the E10.5 anterior trunk, the interactions of the active *Hoxd4* gene correlated primarily with H3K4me3 marks. The contacts established by *Hoxd9* correlated both with H3K27me3 and H3K4me3, either in E8.5 tailbuds, or in E10.5 anterior trunk, likely due to the presence of both expressing and non-expressing cells. At both stages where *Hox* clusters are partially active, the patterns of 3D compartmentalization and histone marks thus strongly correlated. Therefore, step wise *Hox* gene transcriptional activation, at least for the *Hoxd9* to *Hoxd13* genes, is accompanied by a conformational separation between active and inactive domains, which pre-figures their 3D organization at later developmental stages along the AP-axis (*Figure 2*; *Noordermeer et al., 2011*).

## Posterior *Hoxd* genes switch autonomously between 3D compartments

Temporal colinearity was initially defined as the sequential activation of *Hox* genes according to their positions in the clusters (*Izpisua-Belmonte et al., 1991*; *Duboule, 1994*). However, studies on the global transcriptional organization of the *HoxD* cluster, at least in the developing spinal cord, revealed two large and regulatory-independent modules, which separate 'posterior' genes (the *AbdB*-related *Hoxd9* to *Hoxd13* genes) from the rest of the gene cluster (*Tschopp et al., 2012*). Also, in different developmental contexts

Table 1. Continued

| 4C-seq | ChIP-seq | | |
| | Input | H3K27me3 | H3K4me3 |
| --- | --- | --- | --- |
| *Hoxb9* E10.5 Forebrain 2 | 0.03 | **0.59** | 0.16 |
| *Hoxb9* E10.5 Anterior trunk | 0.06 | −0.01 | **0.69** |
| *Hoxa13* ES cells 1 | 0.10 | **0.52** | 0.14 |
| *Hoxa13* E8.5 PSM | 0.10 | **0.58** | 0.12 |
| *Hoxa13* E10.5 Forebrain 1 | 0.07 | **0.60** | 0.22 |
| *Hoxa13* E10.5 Anterior trunk | 0.06 | **0.73** | 0.20 |

Spearman's rank correlation coefficient between pairs of 4C-seq and ChIP-seq samples in different samples (see section 'Material and methods' for methodology). For each 4C-seq sample, the highest correlating ChIP-seq sample is highlighted in bold.

such as the limbs and the cecum, groups of neighboring *Hoxd* genes are activated as single regulatory blocks (*Montavon et al., 2011*; *Andrey et al., 2013*; *Delpretti et al., 2013*). We thus assessed whether the transition in chromatin domains also occurred stepwise or, alternatively, if large domains consisting of multiple genes were initially organized in space, followed by sequential gene transcription within these domains.

We first compared the 3D cluster architecture over the course of embryonic development, between the E8.5 PSM and dissected E9.5 tail buds (*Figure 3*). This latter sample was obtained after cutting off the most caudal part of E9.5 embryos (Theiler stage 15) right after the incipient hind limb bud, that is at ca. somite 26–27 level. Accordingly, this sample contained the tail bud proper as well as some tissue localized slightly more rostral. During this 24 hr time interval, the *Hoxd10* and *Hoxd11* genes become robustly activated in these cells, which are derived from a subpopulation of the sample dissected at E8.5. In the E8.5 PSM, *Hoxd10* and *Hoxd11* are still silenced.

The re-organization of compartmentalization occurring along with gene activation over this 24 hr period was clearly revealed by comparing the profile obtained when using *Hoxd11* as a viewpoint with those obtained with either *Hoxd4* or *Hoxd13* (*Figure 3*, top ratio between E8.5 PSM and E9.5 Tail bud). When switching from an inactive to an active state, *Hoxd11* re-deployed its interactions from the inactive, centromeric compartment (*Figure 3*, top ratio: blue shaded area) to the active telomeric compartment (*Figure 3*, top ratio: brown shaded area). These negative and positive compartments can be identified by the interaction profile of either *Hoxd13* (*Figure 3*, top left) or *Hoxd4* (*Figure 3*, top right), respectively. Accordingly, *Hoxd4* shifted its interactions towards the centromeric (active) part of the cluster in both E9.5 and E10.5 samples (*Figure 3*; right), to contact *Hoxd10*, *Hoxd11* and, to some extent, *Hoxd12* (*Figure 3*, top ratio: brown shaded area). Of note, the dissected E8.5 PSM contained a mixture of cells either positive or negative for *Hoxd9* transcription, which coincided with this gene showing conspicuous contacts with both extremities of the gene cluster, depending whether it was active (right) or inactive (left) (*Figure 3*, top, black arrows). In contrast, the E9.5 dissection contains a more homogenous cell population, strongly expressing this gene. As a consequence, in this latter sample, *Hoxd9* contacts more strongly the now expressed *Hoxd10* and *Hoxd11* genes, whereas the interactions with *Hoxd13* or *Evx2* are strongly diminished (*Figure 3*; compare top with middle panels).

In the E10.5 tail bud, the terminal part of the cluster containing *Hoxd12* and *Hoxd13* has been activated, as shown by the increased contacts established by *Hoxd13* with the telomeric part of the cluster, indicating that the full *HoxD* array had been processed and that all genes were now (at least in part) included into a 'positive' compartment (*Figure 3*, bottom ratio: purple shading). While considerably increased, contacts of *Hoxd13* with the telomeric part of the cluster were however weak (*Figure 3*, bottom, black arrows), most likely reflecting the restricted expression of *Hoxd13* at this stage, in a small subset of the dissected cells. In the same context, contacts established both by *Hoxd4* and *Hoxd9* with the centromeric part of the cluster extended towards the end of the cluster along with the developmental stage, such that both reached *Hoxd12* in E10.5 samples, whereas *Hoxd13* was only weakly contacted, corresponding to the results obtained when using *Hoxd13* as a bait (*Figure 3*; bottom left).

Therefore, it appears that the colinear time sequence in *Hox* genes activation is paralleled by a progressive transition in the chromatin structure, with a positive domain gaining in size along with time, at the expense of the negative domain, as best seen by the extension of *Hoxd4* contacts. At E8.5, these interactions extended up to *Hoxd8-Hoxd9*. In E9.5 samples *Hoxd10* was clearly contacted, and in E10.5 *Hoxd11* and *Hoxd12* were also involved (*Figure 3*, right column). These dynamic

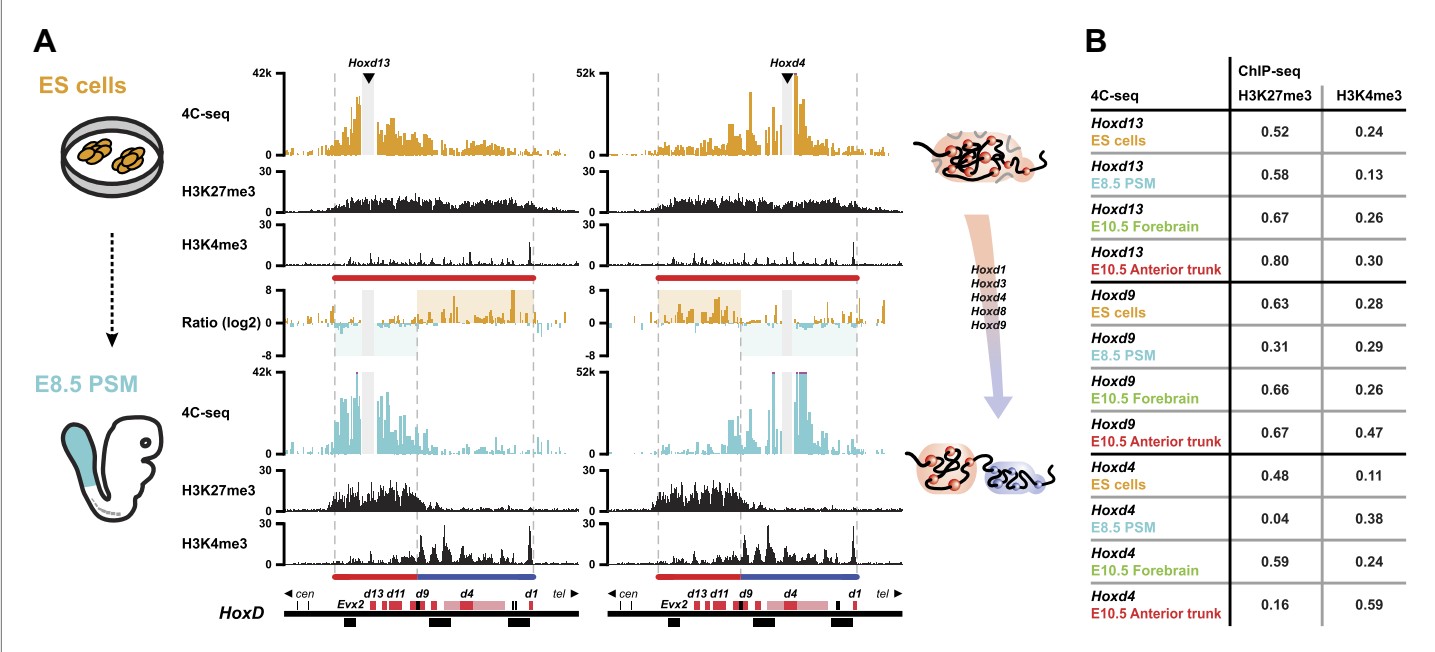

**Figure 2**. Bi-modal 3D organization of *Hox* clusters upon sequential activation. (**A**) Quantitative local 4C-seq signal for the *Hoxd13* (left, centromeric side of *HoxD* cluster) and *Hoxd4* (right, telomeric side of *HoxD* cluster) viewpoints, either in ES (orange), or E8.5 pre-somitic mesoderm (cyan) cells. Below, the H3K27me3 and H3K4me3 ChIP-seq profiles are aligned. The colinear expression status of *Hoxd* genes in each sample is schematized below the ChIP-seq profiles, with active genes in blue and inactive genes in red. Ratios between the 4C-seq signals in different samples are indicated between the profiles. The boundaries separating active from inactive *Hox* gene compartments are indicated by dashed lines. The locations of *Hoxd* genes (red) and other transcripts (black) are shown below. The samples are shown on the left and cartoons summarizing the genome organizations are indicated on the right. (**B**) Spearman's rank correlation coefficient between pairs of 4C-seq and ChIP-seq samples, in early and late embryonic material.
The following figure supplements are available for figure 2:

**Figure supplement 1**. Upon sequential activation, the *HoxD* cluster adopts a bi-modal 3D organization.

**Figure supplement 2**. Upon sequential activation, other *Hox* clusters adopt a bi-modal 3D organization as well.

topologies suggest a stepwise transition of the genes from the negative to the positive compartment, rather than the switch of large groups of multiple transcription units, following a discrete and global chromatin re-organization.

## Memorizing bimodal chromatin configurations

During axial extension, *Hox* genes are activated in the most posterior aspect of the elongating embryo (***Deschamps and van Nes, 2005***). It is thus possible that cells implementing this stepwise transition in chromatin domains can fix and memorize their bimodal distribution once they exit the posterior zone of activation, leading to the colinear *Hox* conformations observed along the AP-axis (***Noordermeer et al., 2011***). Accordingly, one would expect cellular territories along the developing body axis to maintain the same bimodal combinations as those established at the time of their origin, during early axial extension. We looked at the similarities in bimodal profiles between posterior samples dissected at different times on the one hand, and various samples micro-dissected at different body levels, from E10.5 embryos, on the other hand (***Figure 4***).

The profile obtained from E8.5 PSM (***Figure 3***, top), right at the onset of *Hoxd9* activation globally aligned with that observed in the 'anterior trunk' sample at E10.5 (***Figure 4***, top), that is a cellular domain with a posterior boundary positioned approximately at the *Hoxd9* anterior limit of expression. In both cases, *Hoxd9* clearly contacted the negative domain, as defined by the *Hoxd13* contacts (***Figure 4***, top), whereas some weak contacts were also scored with the positive domain, as determined by *Hoxd4*

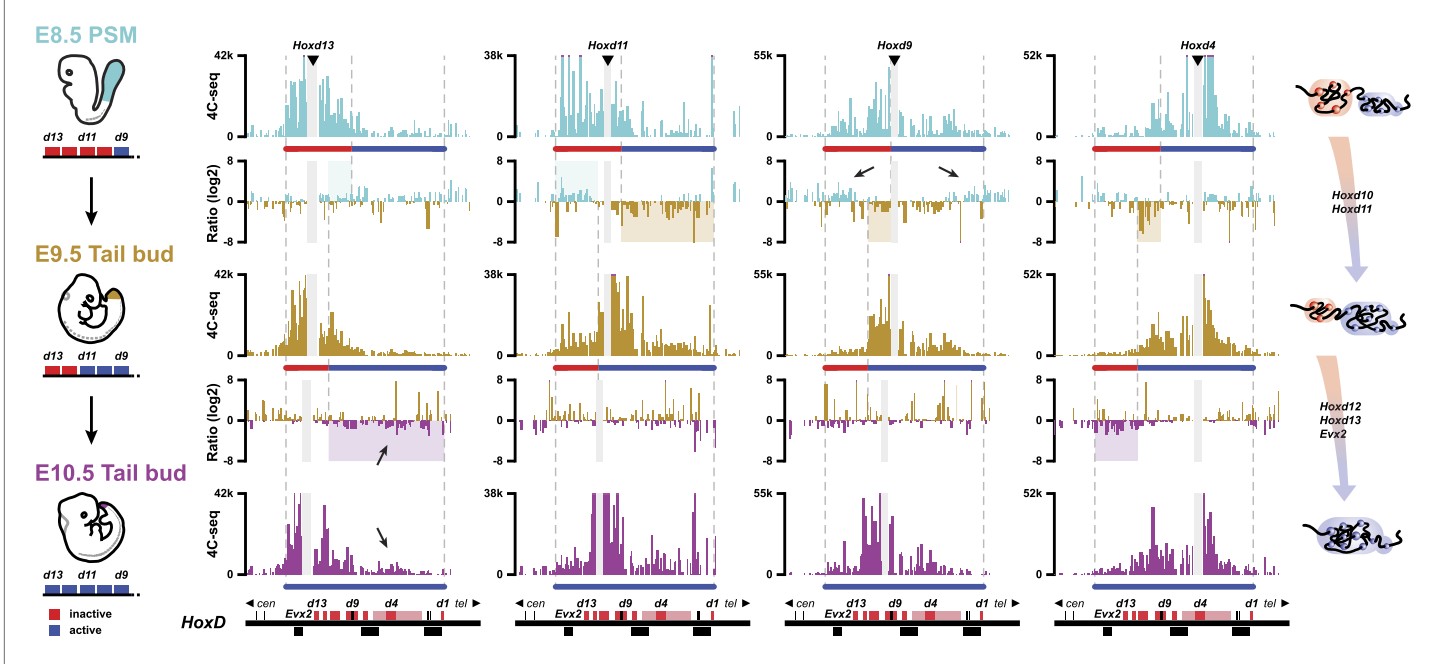

**Figure 3**. Activated *Hoxd* genes switch compartments. Quantitative local 4C-seq signals for the *Hoxd13*, *Hoxd11* *Hoxd9* and *Hoxd4* viewpoints in either E8.5 pre-somitic mesoderm (cyan), E9.5 tail bud (brown) or E10.5 tail bud (purple) cells. The colinear expression status of *Hoxd* genes is schematized below each profile and, on the left, below each cartoon. Ratios between 4C-seq signals in different samples are indicated between the corresponding profiles. The boundaries between active and inactive *Hox* gene compartments are indicated by dashed lines and regions displaying important changes in interactions, as discussed in the text, are highlighted. Black arrows point towards opposing interacting behaviors due to the heterogeneous activity state of the viewpoint in the sample. The locations of *Hoxd* genes (red) and other transcripts (black) are shown below.

contacts, indicating that the posterior limit of the dissection was slightly below the *Hoxd9* boundary. These contacts were somehow stronger, in proportion, in the E8.5 than in the E10.5 dissection.

At E9.5 (Theiler stage 14), the 'tail bud' (i.e., from the start of the non-segmented mesoderm) was dissected from ca. somite 22 to 25 and caudally. At this stage, *Hoxd9* is robustly transcribed, whereas *Hoxd11* has just started transcriptional activation. The interaction profiles obtained in this tissue were most similar to those obtained when a fragment of E10.5 trunk was dissected out that grossly corresponded to the future lumbo-sacral region, at levels 22 to 28, that is the AP levels supposedly produced in the E9.5 tail bud (*Figure 4*, middle). At this AP level, *Hoxd9* is fully activated and this was reflected by the quasi absence of contact with *Hoxd13* whereas, conversely, strong interactions appeared with the active part of the cluster (*Figure 4*, top ratio: red and blue shading). This was controlled by using *Hoxd4* as bait, since contacts were now clearly scored with *Hoxd9* and *Hoxd10* and, to a lesser extent, with *Hoxd11* (*Figure 4*, top ratio: blue shading).

In this lumbo-sacral sample, neither *Hoxd12* nor *Hoxd13* are as yet transcribed, which coincided with the absence of contact between *Hoxd13* and the active part of the gene cluster (*Figure 4*, middle, left). On the other hand, *Hoxd11* expectedly displayed a mixed interaction profile, contacting both the negative and positive domains, likely reflecting the presence of both expressing and non-expressing cells (*Figure 4*, middle). In the most caudal piece of the E10.5 mouse embryo, interactions between *Hoxd12*, *Hoxd13* and the positive domain were finally detected, suggesting that the entire cluster falls into a single spatial domain (*Figure 4*, bottom ratio: purple shading). Here again, however, though the interactions were significant, they were not particularly strong, suggesting the presence of a mixed cell population. Based on these data, we propose that the bimodal distributions are frozen in those cells leaving the zone of proliferation, at the caudal aspect of the embryo where temporal colinearity is potentially processed. These 3D structures, and hence the *Hox* transcription programs, will thus be maintained and memorize the various AP levels from which they originate.

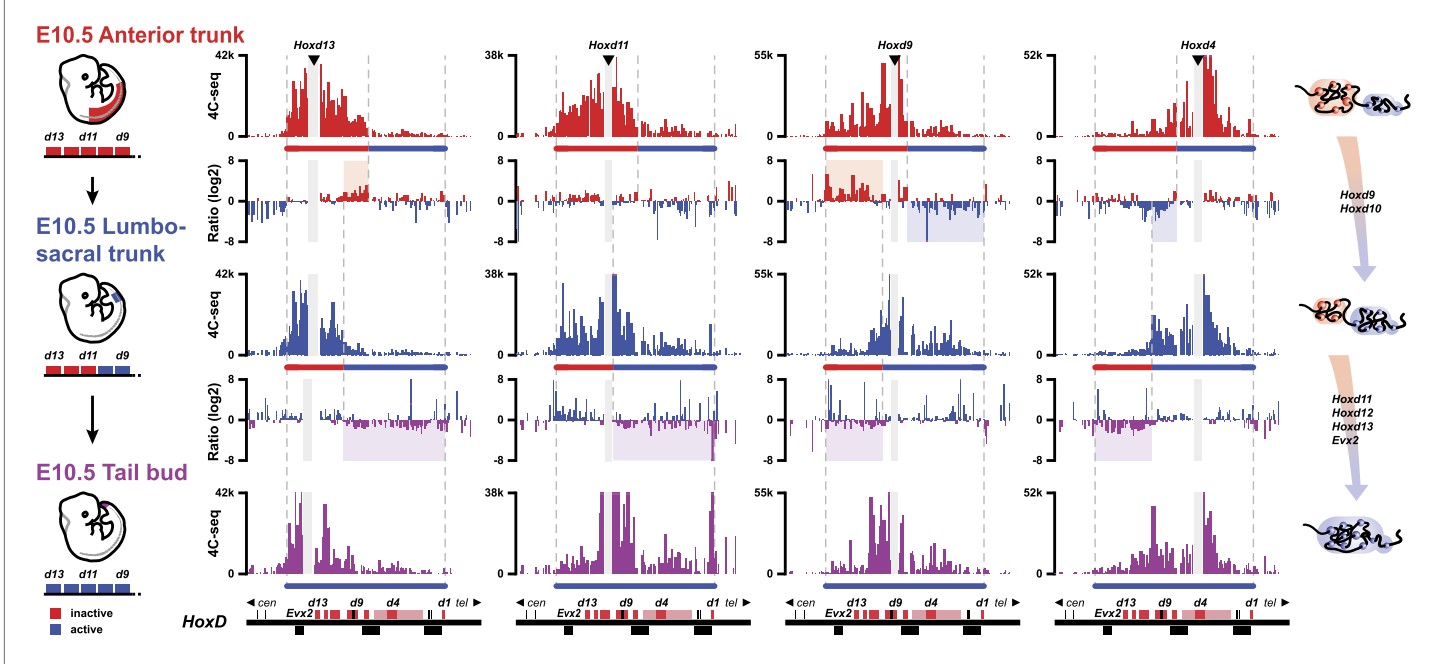

**Figure 4**. The bimodal 3D organization of *Hox* cluster may help memorize states of colinear expression. Quantitative local 4C-seq signals for the *Hoxd13*, *Hoxd11* *Hoxd9* and *Hoxd4* viewpoints, in samples taken at various anterior to posterior positions along the developing body axis from E10.5 embryos. Anterior trunk (red), lumbo-sacral trunk (blue) and tail bud (purple) tissues were used and the approximate expression status of *Hoxd* genes in every sample is schematized below each profile (as for *Figure 3*). Ratios between 4C-seq signals in the different samples are indicated between the corresponding profiles. The boundaries between active and inactive *Hox* gene compartments are indicated by dashed lines and regions displaying important changes in interactions, as discussed in the text, are highlighted. The locations of *Hoxd* genes (red) and other transcripts (black) are shown below. On the right, cartoons summarizing the 3D genome organization of the *HoxD* cluster are indicated.

## Temporal colinearity within a constitutive framework of long-range interactions

In different developmental contexts, the transcriptional activity of *Hoxd* genes coincides with an overall remodeling of long-range chromatin interactions with the flanking gene deserts, which harbor essential enhancer elements active in these developing tissues (*Montavon et al., 2011*; *Andrey et al., 2013*; *Berlivet et al., 2013*; *Delpretti et al., 2013*). Colinear activation of *Hoxd* genes along the developing trunk is thought to primarily rely on regulatory influences intrinsic to the gene cluster itself (*Spitz et al., 2001*). However, and even though their importance remains unclear, contributions of the flanking regulatory landscapes in this process have been proposed (*Tschopp et al., 2009*; *Tschopp and Duboule, 2011b*). Therefore, we assessed whether or not the reported changes in local interactions are associated with variations in long-range contacts during temporal colinearity, as was observed during limb and intestinal development (*Montavon et al., 2011*; *Andrey et al., 2013*; *Delpretti et al., 2013*).

By using a recently developed analytical methodology (*Woltering et al., 2014*), we found that all interrogated *Hoxd* genes displayed substantial interactions with the flanking gene deserts (*Figure 5A*, *Figure 5—figure supplement 1A*). The quantification of interactions over both the centromeric and telomeric gene deserts revealed a gene-specific interaction preference towards either one or the other desert (*Figure 5B*), similar to what was previously described in limb bud cells (*Montavon et al., 2011*; *Andrey et al., 2013*). However, in marked contrast, the dynamics of these long-range chromatin interactions were moderate, if any, and no clear modification in the contact profiles were detected between the inactive state in ES cells, and the subsequent transcriptional activation (*Figure 5B,C*). Hierarchical clustering of global patterns of long-range interactions revealed that the *Hoxd4*, *Hoxd9* and *Hoxd11* viewpoints systematically cluster together, whereas the *Hoxd13* viewpoint always behaves as outlier (*Figure 5—figure supplement 1B*).

This clustering of interactions matches with the position of a previously mapped boundary between 'topological associated domains' (TADs; *Dixon et al., 2012*; *Nora et al., 2012*). In ES cells indeed, two

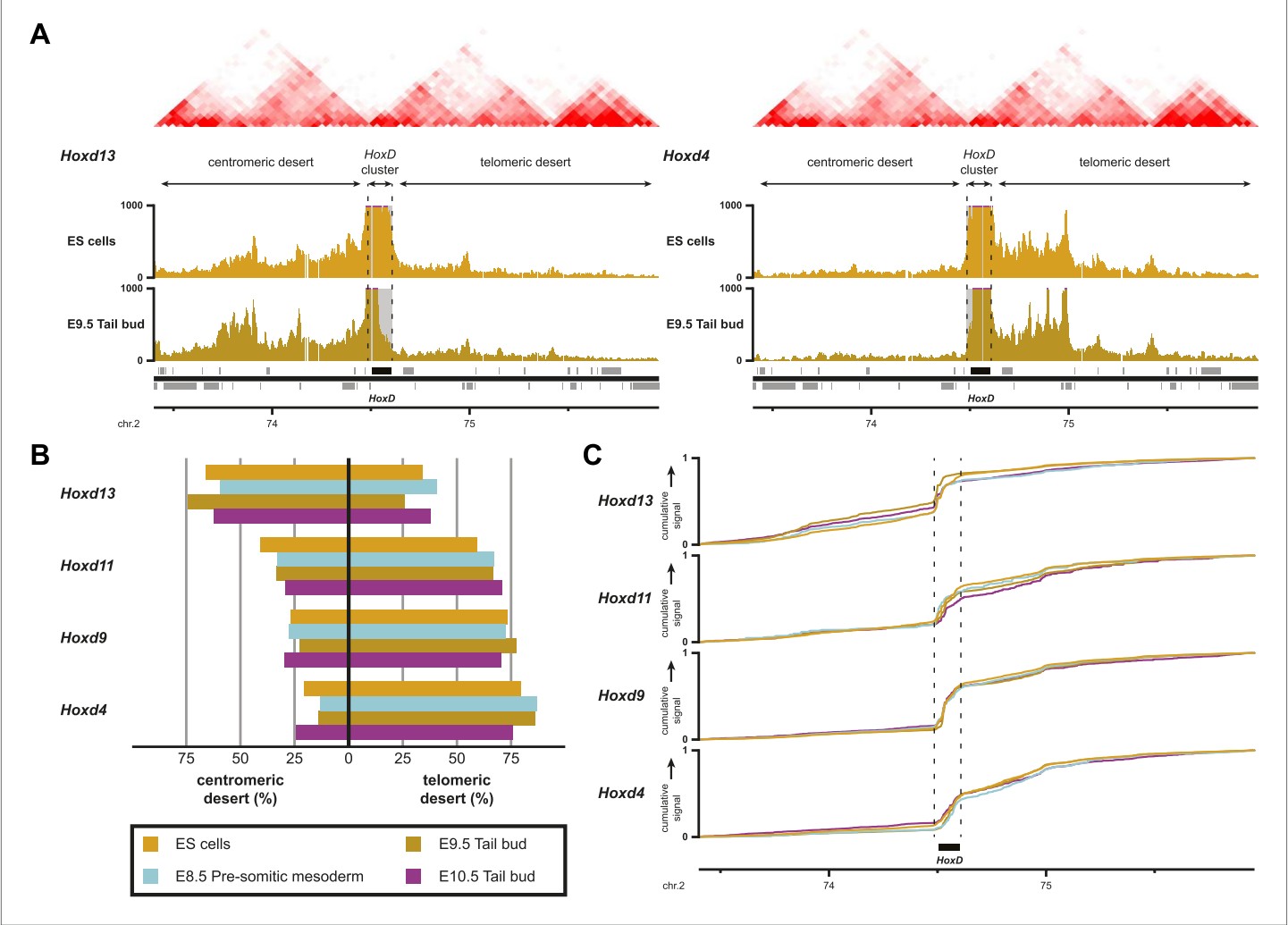

**Figure 5**. Sequential *Hoxd* gene activation occurs without drastic remodeling of long-range interactions. (**A**) Distribution of long-range contacts in both the centromeric and telomeric gene deserts surrounding the *HoxD* cluster. Smoothed 4C-seq signals (11 fragment window size) are shown for the *Hoxd13* and *Hoxd4* gene viewpoints in ES and E9.5 tail bud cells. The analyzed genomic interval is the same as in *Woltering et al. (2014)*. The location of topological domains (TADs) in ES cells are obtained from *Dixon et al. (2012)* and indicated on the top with the *HoxD* cluster and both the centromeric and telomeric gene deserts indicated by arrows. The dashed lines demarcate the domain of high signal over the *HoxD* cluster, which is excluded from the analysis. (**B**) Summaries of the distributions in long-range signals within the centromeric and telomeric gene deserts surrounding the *HoxD* cluster, for all *Hoxd* genes assayed at various stages of their sequential activation. Each *Hoxd* gene specifically interacts with either the centromeric or the telomeric gene desert and these privileged contacts remain largely invariant during transcriptional activation. (**C**) Cumulative signals over the centromeric and telomeric gene deserts and the *HoxD* cluster for all *Hoxd* genes assayed at various stages of their sequential activation.

The following figure supplements are available for figure 5:

**Figure supplement 1**. Temporal colinearity occurs without dynamic long-range interactions.

**Figure supplement 2**. Comparison between HiC and 4C-seq datasets obtained in ES cells.

TADs cover approximately the gene deserts on either side and have their border at the level of the *Hoxd12-Hoxd11* genes (*Dixon et al., 2012*; *Figure 5A*, *Figure 5—figure supplement 1A*). Virtual 4C from HiC data with bins that cover the *Hoxd* genes show highly similar priming of interactions with the surrounding TADs when compared to our 4C-seq analysis, confirming that both approaches score similar chromatin behavior (*Figure 5—figure supplement 2*; *Sexton et al., 2012* for analysis strategy). During limb development, genes located near this boundary (*Hoxd9* to *Hoxd11*) change their tropism and switch their contacts from one TAD to the other, such as to interact sequentially with the appropriate

enhancers (*Andrey et al., 2013*). This structural re-organization is clearly illustrated in our hierarchical clustering with *Hoxd11* changing its association from *Hoxd4* to *Hoxd13* (*Figure 5—figure supplement 1C*). During temporal colinearity, however, such structural re-organization is not observed and hence the stepwise transcriptional activation of *Hoxd* genes appears to occur within a largely constitutive framework of long-range interactions; genes up to *Hoxd11* interact mostly with the telomeric domain, either before or after their activation, and *Hoxd13* always interacts with the centromeric domain. This opposed tropism for *Hoxd* genes in ES cells, as revealed by HiC and the consequent TAD structures, is somewhat at odds with the local clustering of *Hoxd* genes when in a negative state, which we report here by using 4C. This paradox is discussed below.

## Discussion

In this study we present the dynamics of local and long-range 3D chromatin organization during temporal colinear activation of *Hox* genes in vivo. Prior to their sequential activation, *Hox* genes are organized into local 3D chromatin compartments that encompass all bivalently marked chromatin (*Bernstein et al., 2006*). In ES cells, however, these 3D compartments appear less defined than the fully inactive and H3K27me3-only marked compartments observed in differentiated cells. This difference in discreteness was paralleled by the amount of observable background transcription, which was substantially higher in ES cells that in brain cells, though the biological impact of these very low level transcripts, if any, remains to be determined. This supports the view whereby a generally relaxed chromatin organization in ES cell may accompany the plasticity required for cell-faith commitment (*Mattout and Meshorer, 2010*). Interestingly, in ES cells grown in the presence of two kinase-inhibitors and thought to be in a somehow more naïve developmental state, H3K27me3 marks are almost absent from *Hox* clusters (*Marks et al., 2012*). Considering the high correlation between the presence of H3K27me3 marks and the existence of 3D chromatin compartments, we would anticipate *Hox* clusters within these ES cells not to group into 3D compartments as distinct as those visible in 'canonical' ES cells. We therefore hypothesize that structuring into fully inactive 3D compartments is a gradual process occurring over the course of several days during embryonic development.

### The spatial dynamics of temporal colinearity

We also observe that the sequential transcriptional activation of *Hox* genes in the PSM coincides with a gene-by-gene transfer or positioning from the inactive H3K27me3-decorated compartment to a newly formed 3D compartment containing active genes only. This indicates that the presence of *Hox* genes in 3D compartments of various extents, along the developed body axis (*Noordermeer et al., 2011*) is not an a posteriori mechanism used to fix and secure the long-term maintenance of various states of activity, fixed earlier by 'classical' transcriptional regulations acting *in trans*. Instead, it suggests that such spatial structures are instrumental in the precise regulation of their transcriptional timing. This is observed at least for the *Hoxd9* to *Hoxd13* genes and we infer that the same process occurs in the part of the cluster containing from *Hoxd1* to *Hoxd8*. It is however not possible to assess this experimentally due to technical limitations associated with the size of the embryonic material at the corresponding developmental stages.

Temporal colinearity was originally proposed as a mechanism to translate time into spatial coordinates, in different ontogenic contexts (*Duboule, 1994*; *Gerard et al., 1997*; *Durston et al., 2012*). While our results support this idea, the transcriptional timing associated with gene clustering may not be an absolute prerequisite to achieve the proper spatial distributions of *Hox* genes products, as suggested by the multiple cases where single mammalian *Hox* transgenes could largely recapitulate the major expression specificities along the AP-axis (*Krumlauf, 1994*; *Duboule, 2007*; *Tschopp et al., 2009*; *Tschopp and Duboule, 2011a*). In this context, it is possible that the progressive transition of *Hox* genes from an inactive to an active 3D compartment reflects the existence of a mechanism whose major aim would not be to precisely regulate a time sequence but instead, to protect the most 'posterior' *Hox* genes from a premature exposure to activating factors, a situation shown to block posterior elongation and hence to be detrimental to the embryo (*Young et al., 2009*; *Mallo et al., 2010*). The necessity to actively prevent the most posterior genes from premature activation is supported by their basal transcriptional activity in ES cells, where *Hox* clusters are less discrete than in subsequent negative tissues such as fetal brain cells. Such a basal activity, which was not scored in these latter cells, may reflect the rather generic nature of the activating signals, the general mechanism underlying temporal colinearity thus relying on de-repression.

Studies using internal *Hox* cluster deletions and duplications indeed showed that the relative position of *Hox* genes, rather than their promoters, determines their responses to activating signals (*Tschopp et al., 2009*). In this view, graded signals emanating from the posterior aspect of the developing embryo would lead to a progressive de-repression of *Hox* clusters, implying that these clusters would display some directional sensitivity. While the nature of the activating factors is elusive, a link with the segmentation clock was proposed (*Dubrulle et al., 2001*; *Zakany et al., 2001*). Concerning the directional sensitivity, *Polycomb* group (Pc-G) gene products may play an important role in this process, as the distribution of H3K27me3 marks correlate with the size of the inactive 3D compartments. Recently, a somewhat graded distribution of both EZH2 and RING1B, two proteins members of the PRC2 and PRC1 complexes, respectively, was described over the *HoxD* cluster in ES cells, with the highest signals covering the most 'posterior' genes (*Li et al., 2011*). Directionality may therefore derive from a weaker 'anterior' repression exerted by the Pc system. In this context, progressive alterations of the repressive system should sensitize the transcriptional threshold, while keeping on with directionality. This effect was observed in *Cbx2*−/− mutant embryos (a component of the PRC1 complex formerly known as M33), where the efficiency of the PRC1 complex was moderately decreased: RA treatment resulted in premature yet colinear activation of *Hoxd* genes (*Bel-Vialar et al., 2000*). Alternatively, colinear activation may rely upon a different kind of model involving for example biophysical forces (*Almirantis et al., 2013*). Future experiments where the process will be witnessed at the cellular level in real time may be informative in this context.

## Transcriptional maintenance

*Hox* genes are originally activated in the most posterior aspect of the gastrulating embryo. This initial wave of activation seems to involve first a poised transcriptional status (*Forlani et al., 2003*), followed by an apparent anterior forward spreading (*Deschamps and Wijgerde, 1993*; *Gaunt and Strachan, 1994*; *Gaunt, 2001*), which will ultimately lead to the positioning and initiation of the expression domains in the pre-somitic mesoderm (PSM). The colinear processing of this early phase may involve preparatory modifications in the chromatin status, making the system poised for activation by factors emanating from posterior cells (*Forlani et al., 2003*). In this view, the observed anterior forward spreading in expressing cells (*Deschamps and Wijgerde, 1993*; *Gaunt and Strachan, 1994*; *Gaunt, 2001*) may reflect a prolonged exposure to low levels of signals diffusing from the posterior end of the primitive streak (*Forlani et al., 2003*). A second (non-exclusive) possibility is that it illustrates the initial difficulty to maintain a robust boundary in Pc repression in a gene cluster where some anterior genes are fully active, with a tendency for the nearby-located genes to be de-repressed and activated.

However, our results suggest that once the expression is finally established within the PSM, the boundary between the active and inactive compartments remain rather stable for the next couple of days, until the axial skeleton is fully determined. In this view, these chromatin domains may represent part of the machinery used to fix a given state of activation and thus translate a temporal parameter into spatial coordinates. As such, early heterochronies in *Hox* gene activation within the PSM will lead to subsequent re-positioning of the expression boundary, as previously observed (*Gerard et al., 1997*).

## Long-range contacts

By using genetic approaches, it was previously argued that the time-sequenced activation of *Hoxd* genes primarily uses regulatory influences located within the gene cluster itself (*Spitz et al., 2001*), with some contributions coming from more distant flanking regions (*Tschopp et al., 2009*; *Tschopp and Duboule, 2011b*). We now report that such a transcriptional activation is implemented with little-if any-differences in the interaction profiles between the target genes and their neighboring gene deserts, unlike the situation observed during limb development where new contacts appear upon gene activation (*Montavon et al., 2011*; *Andrey et al., 2013*). However, temporal colinearity does occur within a framework of constitutive long-range interactions, which may provide a scaffold helping the bimodal separation of active and inactive genes to take place. Further experiments with mice carrying large re-arrangements of these two gene deserts will be necessary to clearly weight the importance of flanking regions in the implementation of the *Hox* clock.

Finally, while the comparison between published HiC data (*Dixon et al., 2012*) and our 4C datasets are generally highly consistent (e.g., *Figure 5*, *Figure 5—figure supplement 1*; *Andrey et al., 2013*), the data reported here using ES cells raise an apparent paradox. HiC analysis in ES cells identified a boundary between topological domains positioned around the *Hoxd12* to *Hoxd11* gene (*Dixon et al.,*

*2012*; *Figure 5—figure supplement 2D*) and such boundaries are thought to impose or reflect a physical separation between the two interaction landscapes (e.g., *Nora et al., 2013*). As a consequence, *Hoxd13* should display more interactions with its flanking gene desert than with the other part of the *HoxD* gene cluster. Yet, by using several viewpoints in a 4C set-up, the *HoxD* cluster in ES cells appears to form a single negative compartment, despite the interspersed presence of this TAD boundary (*Figure 1*). In fact, a detailed analysis of the HiC dataset reveals that the *HoxD* cluster itself forms a 'micro-TAD', displaying strong internal interactions, in agreement with the 4C results reported here. As such, we consider it likely that the TAD boundary identified by HiC in ES cells (*Dixon et al., 2012*) represents an average description of two distinct configurations (*Figure 6*). For each allele, either the most posterior *Hoxd13* gene forms stable interactions within the TAD on the centromeric side or, alternatively, the *Hoxd11* to *Hoxd1* genes interact with the TAD on the telomeric side (*Figure 6*). As a consequence, for each allele the entire *HoxD* 3D compartment becomes located towards a single TAD, on one side of the cluster with a physical separation from the other side. Molecule(s) causing these interactions are elusive and may include proteins that mediate constitutive loops between *Hoxd* gene promoters and their regulatory elements. The CTCF protein, which may play a role in scaffolding TADs, binds multiple sites around the *Hoxd13* to *Hoxd8* region (*Ferraiuolo et al., 2010*; *Phillips-Cremins et al., 2013*). Because formaldehyde crosslinking has a very short range of action (*Orlando et al., 1997*), the system generates a graded pattern of 4C and HiC interactions from the location of the actual binding sites. Therefore, while in ES cells and for each allele, TAD borders are likely located at either side of the *HoxD* 3D compartment (*Figure 6A*, black lines), our analysis of a large population of cells reflects the equilibrium that exists between these two situations (*Figure 6E*). At later stages, when the *HoxD* cluster adopts a bimodal 3D organization, the tethering of interactions, as illustrated by the existence of TADs on either side, may help implement the separation between activated and repressed *Hox* genes, thereby potentially reducing deleterious regulatory interferences and premature activation of the most posterior *Hox* genes.

## Materials and methods

### Animal care, tissue sampling, ES cell culture and sample preparation

All experiments were performed in agreement with the Swiss law on animal protection (LPA). Tissue samples were isolated at the indicated time points (with maximum 6 hr delay), with day E0.5 being noon on the day of the vaginal plug. Tissue pieces for 4C-sequencing, ChIP-sequencing, RNA-sequencing and Reverse Transcriptase-qPCR were isolated in PBS and subsequently transferred to PBS supplemented with 10% Fetal Calf Serum. 4C-seq and ChIP-seq material was incubated for 45 min with 1 mg/ml collagenase (Sigma-Aldrich, St. Louis, MO), and 4C-seq material was further made single cell using a cell strainer (BD Falcon).

Mouse ES cells were grown under feeder-free conditions on gelatinized plates in Dulbecco's modified Eagle's medium (DMEM, Life Technologies, Carlsbad, CA) supplemented with 17% fetal calf serum, 1 × non-essential amino acids (Life Technologies), 1 × Pen–Strep (Life Technologies), Sodium Pyruvate (Life Technologies), 0.1 mM β-mercaptoethanol, and 1000 U/ml LIF.

Embryonic 4C-seq samples consisted of pooled material from multiple embryos: 129 embryos for E8.5 pre-somatic mesoderm samples, 196 embryos for E9.5 tail bud, 143 embryos for E10.5 tail bud or E10.5 lumbo-sacral trunks and around 20 embryos for each E10.5 forebrain sample. For embryonic ChIP samples, 50 µg of chromatin was cross-linked at a time, of which 10 µg was used per ChIP. To obtain 50 µg of chromatin, 750 E8.5 pre-somatic mesoderm samples or 10 E10.5 forebrains were pooled. Total RNA from E10.5 forebrain was isolated from single embryos. ES cell 4C-seq and ChIP-seq samples were prepared from samples consisting of 20 million cells. 10 µg of ES cell chromatin was used per ChIP. Total RNA was isolated from 1 million cells.

### 4C-sequencing

4C–seq libraries were constructed as previously described (*Noordermeer et al., 2011*). NlaIII (New England Biolabs, Ipswich, MA) was used as the primary restriction enzyme and DpnII (New England Biolabs) was used as the secondary restriction enzyme. For each viewpoint, a total of 1 µg (E9.5 tail bud, E10.5 tail bud, E10.5 lumbo-sacral trunks, E10.5 forebrain and ES cells) or 50 ng (E8.5 pre-somatic mesoderm) of each 4C-seq library was amplified using 16 individual PCR reactions with inverse primers including Illumina Solexa adapter sequences (primer sequences in *Table 2*). Illumina sequencing was

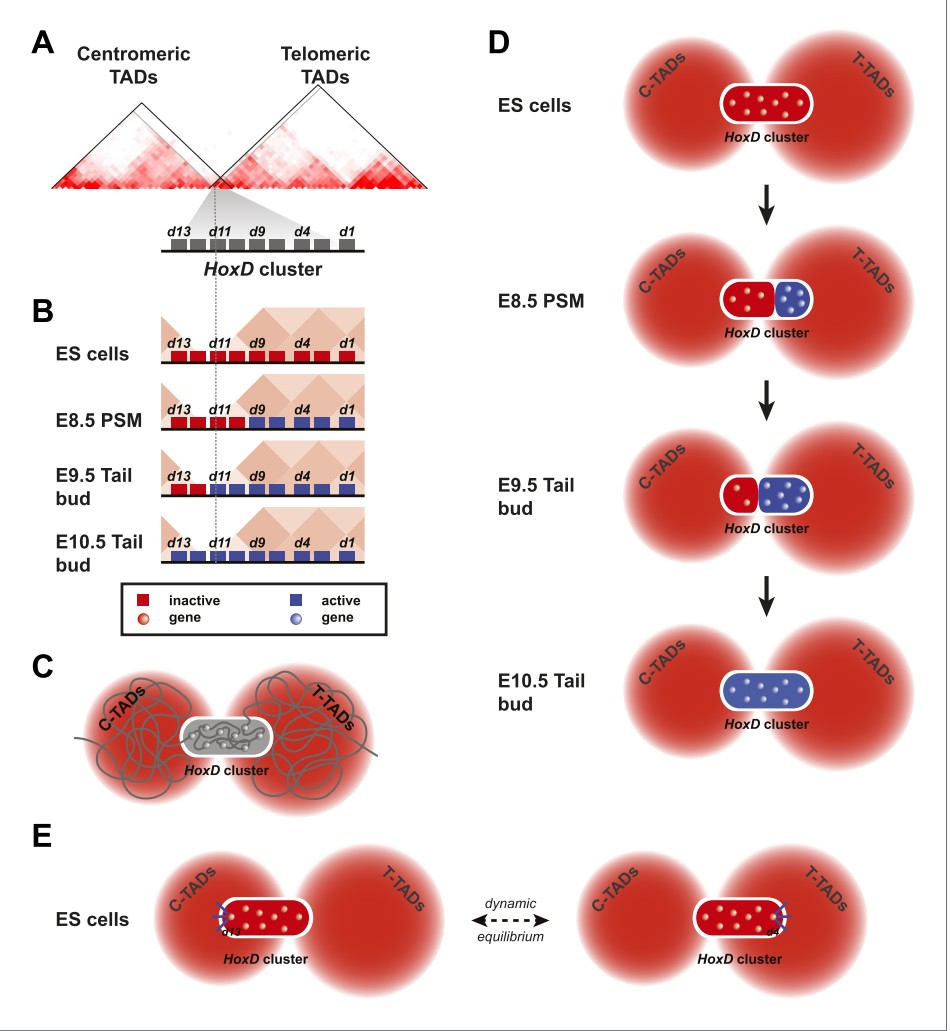

**Figure 6**. Model of dynamic bi-modal 3D compartmentalization during temporal colinearity. (**A**) Schematic organization of topological domains in ES cells (from *Dixon et al. 2012*) matching the centromeric and telomeric gene deserts, with an apparent boundary assigned near the *Hoxd11* gene (grey diagonal lines). All *Hoxd* genes in ES cells have considerable interactions on either side of the cluster, suggesting that this border is more diffuse and hence the entire *HoxD* cluster can be integrated in either TAD (diagonal black lines). (**B**) Various states of activity for *Hoxd* genes in different samples, analyzed during sequential activation. The assigned TAD boundary in ES cells is indicated by the dashed line. (**C**) Conceptual 2D representation of chromatin organization within the *HoxD* cluster chromatin compartment and surrounding centromeric and telomeric TADs in ES cells. (**D**) Schemes illustrating the dynamics of local 3D compartmentalization for the *HoxD* cluster (red and blue compartments) vs the constitutive nature of interactions in the context of the surrounding TADs during sequential activation. (**E**) A dynamic equilibrium to explain the paradox in the observed local vs long-range interactions. Genes located at the centromeric or telomeric extremities of the *HoxD* cluster form stable interactions with DNA sequences located with the flanking gene deserts, thereby dragging the *HoxD* 3D chromatin compartments into either one of the TADs. Within a cellular population, this process is in equilibrium, resulting in a read-out where *Hoxd* genes have a graded preference to interact with either the centromeric or the telomeric deserts, despite being organized into a single 3D chromatin compartment.

done on multiplexed samples, containing PCR amplified material of up to 7 viewpoints, using 100 bp Single end reads on the Illumina HiSeq system according to the manufacturer's specifications.

4C-seq reads were sorted, aligned, and translated to restriction fragments using the 4C-seq pipeline of the BBCF HTSstation (available at http://htsstation.epfl.ch; *Noordermeer et al., 2011*; *David et al., 2014*) according to ENSEMBL Mouse assembly NCBIM37 (mm9). 4C-seq patterns were corrected vs previously generated random 4C–seq libraries (*Noordermeer et al., 2011*), consisting of

**Table 2.** 4C-seq Inverse primer sequences

| Viewpoint | Inverse primer | Sequence |
|---|---|---|
| Hoxd13 | iHoxd13 forward* | AATGATACGGCGACCACCGAACACTCTTTCCCTACACGACGCTCTTCCGATCTAAAAATCCTAGACCTGGTCATG |
| | | chr2:74504328-74504348 |
| | iHoxd13 reverse* | CAAGCAGAAGACGGCATACGAGGCCGATGGTGCTGTATAGG |
| | | chr2:74505579-74505598 |
| Hoxd11 | iHoxd11 forward* | AATGATACGGCGACCACCGAACACTCTTTCCCTACACGACGCTCTTCCGATCTAAGCATACTTCCTCAGAAGAGGCA |
| | | chr2:74523621-74523643 |
| | iHoxd11 reverse* | CAAGCAGAAGACGGCATACGACTAGGAAAATTCCTAATTTCAGG |
| | | chr2:74523881-74523903 |
| Hoxd9 | iHoxd9 forward* | AATGATACGGCGACCACCGAACACTCTTTCCCTACACGACGCTCTTCCGATCTACGAACACCTCGTCGCCCT |
| | | chr2:74536168-74536185 |
| | iHoxd9 reverse* | CAAGCAGAAGACGGCATACGACCCTCAGCTTGCAGCGAT |
| | | chr2:74536797-74536814 |
| Hoxd4 | iHoxd4 forward* | AATGATACGGCGACCACCGAACACTCTTTCCCTACACGACGCTCTTCCGATCTAAGGACAATAAAGCATCCATAGGCG |
| | | chr2:74561330-74561353 |
| | iHoxd4 reverse* | CAAGCAGAAGACGGCATACGATCCAGTGGAATTGGGTGGGAT |
| | | chr2:74562171-74562191 |
| Hoxc13 | iHoxc13 forward* | AATGATACGGCGACCACCGAACACTCTTTCCCTACACGACGCTCTTCCGATCTAGATAATTTTCCTGAGACATTGTAAC |
| | | chr15:102756108-102756132 |
| | iHoxc13 reverse* | CAAGCAGAAGACGGCATACGAGCTCAATGTTCCCTTCCCTAACG |
| | | chr15:102755251-102755273 |
| Hoxb13 | iHoxb13 forward* | AATGATACGGCGACCACCGAACACTCTTTCCCTACACGACGCTCTTCCGATCTAGGACTGTTCCTCGGGGCTAT |
| | | chr11:96057673-96057692 |
| | iHoxb13 reverse* | CAAGCAGAAGACGGCATACGAATCTGGCGTTCAGAGAGGCT |
| | | chr11:96057448-96057467 |
| Hoxb9 | iHoxb9 forward* | AATGATACGGCGACCACCGAACACTCTTTCCCTACACGACGCTCTTCCGATCTAAGATTGAGGAGTCTGGCCACTT |
| | | chr11:96136070-96136091 |
| | iHoxb9 reverse* | CAAGCAGAAGACGGCATACGATCATCAAACCAAGCAGGGCA |
| | | chr11:96136671-96136690 |
| Hoxa13 | iHoxa13 forward* | AATGATACGGCGACCACCGAACACTCTTTCCCTACACGACGCTCTTCCGATCTAACACTTGCACAACCAGAAATGC |
| | | chr6:52212211-52212232 |
| | iHoxa13 reverse* | CAAGCAGAAGACGGCATACGAGGCGAGGCTCAGGCTTTTAT |
| | | chr6:52212476-52212495 |
| CNS(39) | iCNS(39) forward† | AATGATACGGCGACCACCGAACACTCTTTCCCTACACGACGCTCTTCCGATCTATCCAAGGAGAAAGGTGTTGGTC |
| | | chr2:74975258-74975279 |
| | iCNS(39) reverse† | CAAGCAGAAGACGGCATACGACAGGGCGTTGGGTCACTCT |
| | | chr2:74975670-74975687 |

Location of primers according to NCBI37 (mm9).
*Primers from Noordermeer D, Leleu M, Splinter E, Rougemont J, De Laat W, Duboule D. 2011. The dynamic architecture of Hox gene clusters. Science 334:222–225.
Table 2. Continued on next page

*Table 2. Continued*

†Primers from Andrey G, Montavon T, Mascrez B, Gonzalez F, Noordermeer D, Leleu M, Trono D, Spitz F, Duboule D. 2013. A switch between topological domains underlies *HoxD* genes collinearity in mouse limbs. Science 340:1234167.

BACs covering the mouse Hox clusters (*HoxD*: RP23-331E7; *HoxC*: RP23-430C12; *HoxB*: RP23-381I12 and RP23-196F5; *HoxA*: RP24-298M24). After random correction, three restriction fragments were removed that returned aberrant values (*HoxD*: chr2:74'597'000-74'597'732; chr2:74'608'796-74'609'312, *HoxB*: chr11:95'999'958-96'000'916) due to sequence abnormalities in the BAC template (confirmed by Sanger sequencing; not shown). Normalization and further data processing was done as previously described (*Noordermeer et al., 2011*). Quantitative log2 ratios were calculated by dividing the quantitative fragment count between tissue samples. Unprocessed 4C-seq data is available from the Gene Expression Omnibus (GEO) repository under accession number GSE55344. Random corrected tracks are available from http://duboule-lab.epfl.ch/data.

The directionality of long-range interactions was calculated as previously described (*Woltering et al., 2014*). In *Figure 5A*, the smoothed 4C-seq patterns (running mean, window size 11) were obtained using the 4C-seq pipeline of the BBCF HTSstation (available at http://htsstation.epfl.ch; *David et al., 2014*). HiC data on topological associated domains (TADs) from ES cells were obtained from (http://chromosome. sdsc.edu/mouse/hi-c/database.php; *Dixon et al., 2012*). Two TADs located centromeric and telomeric of the clusters were selected, covering genomic coordinates chr2:73400000-75960000 (discussed in *Woltering et al., 2014*). Spearman correlation of long-range patterns was done over the region covering these TADs, with signal on the *HoxD* cluster itself removed (excluded region: chr2:74484971-74607492). Conventional hierarchical clustering was done to score for relationships between viewpoints.

## ChIP-sequencing

ChIP was performed as previously described (*Noordermeer et al., 2011*). Cells were fixed for 5 min in a 2% formaldehyde solution at room temperature. ChIP-seq samples were fragmented to a range of 200–500 bp using tip sonication (Misonix S4000, Misonix, Farmingdale, NY), For all ChIP assays, 10 µg of cross-linked chromatin was used. Antibodies used: anti Histone H3K27me3 (#17-622; Millipore, Billerica, MA) and anti H3K4me3 (#17-614; Millipore). ChIP-seq libraries were constructed from 6 to 10 nanograms of immune-precipitated DNA according to the manufacturers instructions (Illumina, San Diego, CA). Sequencing was done using 50 or 100 bp Single end reads on the Illumina HiSeq system according to the manufacturer's specifications. ChIP-seq reads were mapped to ENSEMBL Mouse assembly NCBIM37 (mm9), and extended to 100 bp if read lengths smaller than 100 bp were used, using the ChIP-seq pipeline of the BBCF HTSstation (available at http://htsstation.epfl.ch; *David et al., 2014*). ChIP-seq data is available from the Gene Expression Omnibus (GEO) repository under accession numbers GSE55344 and GSE31570.

## Correlation of 4C-seq and ChIP-seq samples

Random corrected 4C-seq and ChIP-seq samples were correlated by ranking experimental values within restriction fragments (*Table 1*). First, to each NlaIII restriction fragment covered by the random 4C tracks within the regions visualized in *Figures 1 and 2* (*HoxD* cluster: chr2:74454783-74622413; *HoxC* cluster: chr15:102715179-102909417; *HoxB* cluster: chr11:95992344-96244915; *HoxA* cluster: chr6:52058584-52234371), the average ChIP-seq signal was assigned for each condition. Restriction fragment within individual samples were ranked based on their 4C-seq or ChIP-seq value and subsequently the Spearman's rank correlation coefficient was calculated between pairs of samples.

## RNA-sequencing and Reverse Transcriptase-qPCR

Total RNA from tissue samples was isolated using Trizol LS reagent (Life Technologies). Total RNA from ES cell samples was isolated using Trizol reagent (Life Technologies). For RNA-seq, the RNA was depleted from rRNAs and, subsequently, strand-specific total RNA-seq libraries were constructed according to the manufacturers instructions (Illumina). Sequencing was done using 50 bp Single end reads on the Illumina HiSeq system according to the manufacturer's specifications. RNA-seq reads were mapped to ENSEMBL Mouse assembly NCBIM37 (mm9) and translated into reads per gene (RPKM) using the RNA-seq pipeline of the BBCF HTSstation (available at http://htsstation.epfl.ch; *David et al., 2014*). RNA-seq data is available from the Gene Expression Omnibus (GEO) repository

**Table 3.** RT-qPCR primer sequences

| Fragment | Primer | Sequence |
|---|---|---|
| mRNA | mRNA Tubb2c forward* | GCAGTGCGGCAACCAGAT chr2:25080064-25080081 |
| *Tubb2c* | mRNA Tubb2c reverse* | AGTGGGATCAATGCCATGCT chr2:25079711-25079730 |
| mRNA | mRNA Tbp forward* | TTGACCTAAAGACCATTGCACTTC chr17:15644342-15644365 |
| *Tbp* | mRNA Tbp reverse* | TTCTCATGATGACTGCAGCAAA chr17:15650497-15650518 |
| mRNA | mRNA Hoxd13 forward* | GGTGTACTGTGCCAAGGATCAG chr2:74507077-74507098 |
| *Hoxd13* | mRNA Hoxd13 reverse* | TTAAAGCCACATCCTGGAAAGG over intron boundry |
| mRNA | mRNA Hoxd9 forward* | GCAGCAACTTGACCCAAACA over intron boundry |
| *Hoxd9* | mRNA Hoxd9 reverse* | GGTGTAGGGACAGCGCTTTTT chr2:74537278-74537298 |
| mRNA | mRNA Hoxd4 forward | TCAAGCAGCCCGCTGTGGTC chr2:74565709-74565728 |
| *Hoxd4* | mRNA Hoxd4 reverse | TCTGGTGTAGGCCGTCCGGG chr2:74566355-74566374 |
| mRNA | mRNA Hoxb13 forward | GTCCATTCTGGAAAGCAG chr11:96056334-96056351 |
| *Hoxb13* | mRNA Hoxb13 reverse | AAACTTGTTGGCTGCATACT chr11:96057389-96057408 |
| mRNA | mRNA Hoxb9 forward | GGCAGGGAGGCTGTCCTGTCT chr11:96133282-96133302 |
| *Hoxb9* | mRNA Hoxb9 reverse | GCCAGTTGGCAGAGGGGTTGG chr11:96135938-96135958 |

Location of primers according to NCBI37 (mm9).

*Primers from Montavon T, Le Garrec JF, Kerszberg M, Duboule D. 2008. Modeling *Hox* gene regulation in digits: reverse collinearity and the molecular origin of thumbness. Genes Dev 22:346–359.

under accession numbers GSE55344. For RT-qPCR, cDNA was synthesized after DNAseI treatment (Life Technologies) using SuperScript III (Life Technologies) and oligo-dT primers (Life Technologies), using the manufacturer's instructions. For ES cells and E10.5 forebrain, 2 µg of RNA was used as input for the cDNA synthesis, for E10.5 posterior trunk 1 µg of RNA was used. Products were quantified by qPCR using EXPRESS SYBR GreenER mixes (Life Technologies) on a CFX96 PCR Detection System (BioRad, Hercules, CA). Sequences of intron-spanning primers are provided in *Table 3*.

## Acknowledgements

We thank the members of the Duboule labs in Lausanne and Geneva for useful discussion and are grateful to Joost Woltering for sharing autopod and zeugopod 4C-data. We thank Mylène Docquier, Christelle Barraclough, Céline Delucinge and Natacha Civic from the Geneva Genomics Platform and Jacques Rougemont from the Bioinformatics and Biostatistics Core Facility (BBCF) of the Ecole Polytechnique Fédérale (EPFL) in Lausanne for their assistance in generating and analyzing high throughput data. Computations were performed at the Vital-IT (http://www.vital-it.ch) Center for high-performance computing of the SIB Swiss Institute of Bioinformatics using tools developed by the BBCF (http://bbcf.epfl.ch).

## Additional information

### Funding

| Funder | Grant reference number | Author |
|---|---|---|
| Swiss National Research Foundation | 310030B_138662 | Denis Duboule |
| European Research Council | 232790 | Denis Duboule |

The funder had no role in study design, data collection and interpretation, or the decision to submit the work for publication.

### Author contributions

DN, Conception and design, Acquisition of data, Analysis and interpretation of data, Drafting or revising the article; ML, Conception and design, Acquisition of data, Analysis and interpretation of data;

PS, Acquisition of data, Analysis and interpretation of data; EJ, Conception and design, Acquisition of data; FC, Organized and planified the production of all the mice used, Conception and design; DD, Conception and design, Analysis and interpretation of data, Drafting or revising the article

## Ethics

Animal experimentation: All experiments were performed in agreement with the Swiss law on animal protection (LPA) under license 1008/3482/0 to DD.

---

## Additional files

### Major datasets

The following dataset was generated:

| Author(s) | Year | Dataset title | Dataset ID and/or URL | Database, license, and accessibility information |
|---|---|---|---|---|
| Noordermeer D, Leleu M, Duboule D | 2014 | Temporal dynamics and developmental memory of 3D chromatin architecture at Hox gene loci | GSE55344 | NCBI Gene Expression Omnibus with open, unrestricted access. |

The following previously published dataset was used:

| Author(s) | Year | Dataset title | Dataset ID and/or URL | Database, license, and accessibility information |
|---|---|---|---|---|
| Noordermeer D, Leleu M, Duboule D | 2011 | The Dynamic Architecture of Hox Gene Clusters | GSE31570 | NCBI Gene Expression Omnibus with open, unrestricted access. |

---

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
