## [Decision Letter]

Thank you for sending your work entitled “Temporal dynamics and developmental memory of 3D chromatin architecture at *Hox* gene loci” for consideration at eLife. Your article has been favorably evaluated by a Senior editor and 3 reviewers, one of whom is a member of our Board of Reviewing Editors.

The Reviewing editor (Robb Krumlauf) and the other reviewers discussed their comments before we reached this decision. The combined consensus view is that in principle this is suitable for *eLife* but there are a number of major and minor issues that need to be addressed or considered before it is acceptable for publication in *eLife*. The Reviewing editor has assembled the following comments to help you prepare a revised submission.

In several recent papers this group, as well as others, have shown that the Hox loci are split in active and inactive domains, with these domains associating with either centromeric or telomeric gene desert flanking the locus. The current manuscript largely builds on these earlier findings, confirming that these domains also form during temporal colinearity, and adds more details to this emerging story.

In the current work the authors perform similar 4C studies to their published work, but analyze different tissues. Here the authors focus on exploring how the temporal dynamics of Hox gene activation along the antero/posterior axis correlate with the 3D architecture of the clusters. They show that inactive Hox genes are organized in a single topological domain, as shown in brain and ES cells. However, as the collinear activation of these genes takes place, they move one-by-one, instead of in groups from an inactive to an active bimodal 3D domain. The main result is that active Hox genes occupy the same domain, and have more constant interactions with the gene desert regions. Whether the move is the consequence of gene activation, or is part of the activation process itself remains an open question. These domains coincide, with regions decorated by or depleted of H3K27me3 repression marks, respectively. Interestingly, once this bimodal 3D organization is defined for a particular A/P position, it remains fixed as development progress. Finally, they show that these internal switches within the Hox clusters are independent of the external contact that Hox genes make with the flanking gene deserts, which remain largely invariant through development and correlate with the Topological domains (TADs) defined by HiC studies. This is interesting, as the authors previously showed that these interactions do change and contribute to Hox gene expression during limb development.

Overall the results confirm the previous work, but extend this to a temporal dimension during development. This is a very interesting descriptive work that introduces a temporal dimension to the process of chromatin 3D organization of the Hox loci during collinear activation. There are a number of major and minor issues that need to be addressed or considered before it is acceptable for publication in *eLife*.

*Major*
*comments:*

1) The authors propose an equilibrium model to explain the apparently contradiction between the 4C-seq and the HiC data observed in ES cells. Looking at the 4C-seq data from all other developmental stages, it seems is that an equilibrium between TAD constrains and the Hox interactions within the cluster could be occurring in all tissues and stages. Thus, it seems that, within the Hox clusters (that it, not considering long-range contacts), the architecture restrictions imposed by the TADs are much stronger than the 3D rearrangements caused by the progressive transition from inactive to active domains during A/P specification. Indeed, for the external genes, the increment of the contacts seen during these transitions, although detectable, is rather small compared with the number of interactions identified at their corresponding side of the TAD. It seems that genes closer to the TADs (specially Hox11 genes) are the ones showing the larger changes in their contacts as they transit from an active to an inactive state. This resembles very much with what the authors have shown for long-range contacts of these same genes during the distal limb differentiation, suggesting a more permeability of the TAD border for genes in very close proximity. It would be interesting to plot the number of reads within the cluster that lie at each side of the TADs for each gene and at each stage and A/P position. This could help to determine the putative constrains imposed by TADs onto the 3D dynamics associated to Hox gene collinear activation. This should also be discussed with more detail.

2) Could it be that these strong restrictions imposed by TADs contribute to preventing the premature activation of the most posterior genes along the body axis, which is detrimental for proper development?

3) The authors need to more quantitatively analyze 4C data tracks in order to more rigorously define domains. The work depends on analysis of 4C data. One limitation of this data is that it is less suited for identification of domains than Hi-C, as only single view points are analyzed. Typically topological domains are more readily detectable, both quantitatively and statistically, by Hi-C or 5C, because those methods provide matrices of interaction in which domains are more readily, and rigorously, identified. The authors need to describe their 4C data more quantitatively when they infer domain boundaries: in the current manuscript it seems that domain boundaries are either taken from published Hi-C data (which is from a different tissue/cell type and thus may be incorrect in certain aspects, even though many domain features may be conserved), or are based solely on visual inspection of the data, which is not quantitative. Overall the visual inspection of the data is quite satisfying and domain boundaries are certainly visible yet I think the manuscript would be strengthened by including some statistical metrics to better define a 4C domain. This will also allow more quantitative detection of the precise boundary, which is a key factor in the current work, and would allow a more quantitative comparison with domains of histone modifications. The approach to correlate 4C data with histone modifications described in Table I is a start, but it does not show whether domain boundaries coincide. For instance, 4C signal often appear to extend beyond the domain of histone modification (e.g., Figure 3, Hoxd13 in E9.5 tail bud).

4) Topological domains are much more easily detected by Hi-C. It would be very informative to have high resolution (i.e., deeply sequenced) Hi-C maps for E8.5 PSMs, E10.5 Tail buds, and limb tissues. This would be a more useful Hi-C data set than the ES cells data from Dixon et al. If the authors have this type of analysis they should consider including it in the paper.

5) The section about memorizing chromatin configurations is hard to follow, as it requires visual comparison of 4C data tracks in Figure 4 with those in Figure 3. Can the authors include a supplemental figure that displays these different profiles side-by-side (e.g., the E9.5 Tail bud tracks vs. the E10.5 Lumbo-sacral trum track and include a ratio plot)? The authors also need a more quantitative approach to compare such profiles, as the current comparison is based solely on visual inspections, and important differences between 4C profiles are rather subtle.

6) There are similarities of the model proposed by the authors in Figure 6 and those with the work from Wendy Bickmore's laboratory where they have shown that the relocation of Hox clusters away from the chromosome territory (CT) occurs during their activation, or similarities to Spyros Papageorgiou's “cluster translocation model” hypothesis. In the former, the Hox complex is gradually excluded from CTs, which could correspond to the flanking TADs. In the latter, sequential activation of the complex would involve its translocation from a region of repression (chromatin) to activation where there would be greater access to upstream regulatory factors. Although, Spyros envokes morphogenic and electromotive (i.e., Coulomb force) components, nowhere do the authors mention these works in the content of their discussion or model.

*Minor*
*points:*

1) The statement that ES cells have a less compact Hox locus conformation are difficult to support by just 4C data. FISH might address this if it is available. The authors may wish to soften their conclusion otherwise.

2) The explanation of the apparent paradox between Hi-C and 4C data discussed at the end of the paper is reasonable, but a more likely explanation is that Hi-C simply has a lower resolution (20Kb at best). The authors suggest an apparent paradox when comparing Hi-C data and 4C data: the Hi-C data suggest a TAD boundary around d12 in ES cells, whereas d13 interacts with the rest of the Hox locus as observed by 4C. Their interpretation could be true, but in my view a more simple explanation is that the resolution of Hi-C (20 Kb at best) may be too low, and in the Dixon et al. data the true boundary can easily be 20Kb to the right or the left. I think the argument that Hi-C detects a boundary, while 4C does not, because Hi-C represents the average of two distinct conformations is not valid: 4C should be equally detecting this same average.

3) In ES cells low basal expression of Hox genes is observed, and the locus appears to be in a single domain. The results certainly indicate that any local domain feature may be weaker in these cells as compared to forebrain. However, for the authors to conclude that this reflects a less compact organization they need FISH data. Further, can the authors rule out that the increased outside-the-domain interactions in ES cells are not due to increased random ligation in ES cells?

4) The authors find that genes switch autonomously between compartments. But it seems that d1-d8 act as a single domain, then D9-10-11 get included one by one, and d13 is always different (consistent with it being in a different domain based on Hi-C data). Perhaps the authors can either clarify this, or discuss this in the paper: do they think each gene is acting independently, or are there three groups (D1-D8; D9-11, D13)?

5) The data in Figure 5 is puzzling: Hox d11 is switching domains within the Hox cluster from E8.5 to E10.5, yet it does have the same bias to interact with the telomeric desert as the d4 gene. Does this mean that d4 and d11 are really in the same larger topological domain at all time points, but that there is a change in sub-domain organizations? Cell type specific re-organization of the internal organization of topological domains has been proposed before (e.g., Phillips-Cremins et al. Cell 2013).

6) All results combined, one can imagine a constant global domain organization with temporal changes in sub-domains. It would be very informative to have high resolution (i.e., deeply sequenced) Hi-C maps for E8.5 PSMs, E10.5 Tail buds, and limb tissues. This would be a more useful Hi-C data set than the ES cells data from Dixon et al. I do realize this may be difficult given the amount of material that this may require, but it would be a very interesting dataset as it may reveal the stage-specific sub-domains as well as invariant larger domains in a single dataset, with potential changes in the larger domains in limb.

7) In the legend of Figure 6, the authors indicate that the 'apparent boundary' near the Hoxd11 gene is indicated by grey lines but do they not mean the dashed blue line passing from Figure 6 into 6B? Or are they referring to the grey shaded triangle projecting from the middle of Hoxd11 onto the cartoon of the HoxD complex below it?

8) In regards to the replicates presented in supplemental data to Figure 1, where the contacts within and outside the HoxD complex are represented as a graphical ratio between ES cells and forebrain cells: visually this conveys the difference in the 3D organization of the inactive Hox complex between the ES cells and forebrain cells, however between replicates there is surprising variation in the ratio. Would we not expect less variation in the produced ratios between replicates which are produced from pooled samples of ES cells and forebrain cells? In the absence of the associated chromatin marks, this makes it hard to argue that in ES cells the HoxD cluster is less defined than in forebrain cells. Furthermore, one would worry that it would be easy to pick data from one replicate over another to fit the hypothesis regarding the remodeling of the HoxD architecture during sequential gene activation.

---

## [Author Response]

Major comments:

*1) The authors propose an equilibrium model to explain the apparently contradiction between the 4C-seq and the HiC data observed in ES cells. Looking at the 4C-seq data from all other developmental stages, it seems is that an equilibrium between TAD constrains and the Hox interactions within the cluster could be occurring in all tissues and stages. Thus, it seems that, within the Hox clusters (that it, not considering long-range contacts), the architecture restrictions imposed by the TADs are much stronger than the 3D rearrangements caused by the progressive transition from inactive to active domains during A/P specification. Indeed, for the external genes, the increment of the contacts seen during these transitions, although detectable, is rather small compared with the number of interactions identified at their corresponding side of the TAD. It seems that genes closer to the TADs (specially Hox11 genes) are the ones showing the larger changes in their contacts as they transit from an active to an inactive state. This resembles very much with what the authors have shown for long-range contacts of these same genes during the distal limb differentiation, suggesting a more permeability of the TAD border for genes in very close proximity*.

We do not fully understand this comment. In the limb, *Hoxd11* clearly shifts its contacts from the telomeric domain (long range) to the centromeric domain (long range too).

During trunk development, we show here that this shift does *not* occur and that long-range contacts do not change significantly. Instead, what change are the short-range contacts, with *Hoxd11* shifting from a negative (H3K27me3) ‘micro’-domain to a positive domain including 3’ located genes. This is the main conclusion of the paper in fact, i.e., that the bimodal regulation observed in limbs is NOT at work in the trunk, where a much more progressive mechanism operates, with genes moving one after the other from one domain to the other (and *not* from one TAD to the other...). We have tried to make this point clearer in the text.

*It would be interesting to plot the number of reads within the cluster that lie at each side of the TADs for each gene and at each stage and A/P position. This could help to determine the putative constrains imposed by TADs onto the 3D dynamics associated to Hox gene collinear activation. This should also be discussed with more detail*.

We have done this exercise and we now show the results below (Figure 7). We have added this figure to the rebuttal rather than to the ‘paper’ since the correct interpretation of this calculation is difficult, due to three effects occurring simultaneously (see figure legend), of which the influence of the region flanking the baits, which we exclude in the counting of the reads, can’t be reliably determined. Importantly though, the numbers generally suggest that dynamics within the cluster (even without taking into account changes in the regions flanking the baits) are considerably more prominent that changes in the TADs.Author response image 1.

*2) Could it be that these strong restrictions imposed by TADs contribute to preventing the premature activation of the*
*most posterior genes along the body axis, which is detrimental for proper development*?

Yes, this is an interesting possibility and the last sentence of the Discussion already touched upon this subject. We have tried to make it more explicit and changed it to: “…the tethering of interactions, as illustrated by the existence of TADs on either side, may help implement the separation between activated and repressed *Hox* genes, thereby potentially reducing deleterious regulatory interferences and premature activation of the most posterior *Hox* genes.”

*3) The authors need to more quantitatively analyze 4C data tracks in order to more rigorously define domains. The work depends on analysis of 4C data. One limitation of this data is that it is less suited for identification of domains than Hi-C, as only single view points are analyzed. Typically topological domains are more readily detectable, both quantitatively and statistically, by Hi-C or 5C, because those methods provide matrices of interaction in which domains are more readily, and rigorously, identified. The authors need to describe their 4C data more quantitatively when they infer domain boundaries: in the current manuscript it seems that domain boundaries are either taken from published Hi-C data (which is from a different tissue/cell type and thus may be incorrect in certain aspects, even though many domain features may be conserved), or are based solely on visual inspection of the data, which is not quantitative. Overall the visual inspection of the data is quite satisfying and domain boundaries are certainly visible yet I think the manuscript would be strengthened by including some statistical metrics to better define a 4C domain. This will also allow more quantitative detection of the precise boundary, which is a key factor in the current work, and would allow a more quantitative comparison with domains of histone modifications. The approach to correlate 4C data with histone modifications described in Table I is a start, but it does not show whether domain boundaries coincide. For instance, 4C signal often appear to extend beyond the domain of histone modification (e.g.,*
Figure 3*, Hoxd13 in E9.5 tail bud)*.

We do not fully understand these issues and we feel that the reviewers may have been confused. First, we do not try and define any TAD by 4C in this paper (even though the precision of 4C in defining TAD boundaries by using multiplexed baits is *much* higher than by using 5C or Hi-C – see Andrey et al., *Science*, 2013). We introduce the TAD boundary as defined by Dixon et al. only for a matter of discussion. What we show is precisely that we do not fit a TAD boundary during sequential gene activation. Secondly, in Figure 3 no data on histone modifications is shown and hence we hardly understand what the referee means by: *For instance, 4C signal often appear to extend beyond the domain of histone modification (e.g.,*
Figure 3*, Hoxd13 in E9.5 tail bud).* Instead, we show ratios for the difference in 3D organization between the data sets. In fact, we use these ratio plots to determine the borders between ‘interaction domains’ (not TADs). We consider this controlled approach based on initial normalization and subsequent calculations of ratios (described in detail in the supplemental information of [36]
*Science*) considerably more precise than *‘...based solely on visual inspection of the data’* to determine positions of borders in these highly heterogeneous early embryonic tissue samples. We have tried to improve the quality of the text related to this part to prevent readers being confused.

*4) Topological domains are much more easily detected by Hi-C. It would be very informative to have high resolution (i.e., deeply sequenced) Hi-C maps for E8.5 PSMs, E10.5 Tail buds, and limb tissues. This would be a more useful Hi-C data set than the ES cells data from Dixon et al. If the authors have this type of analysis they should consider including it in the paper*.

We of course agree that HiC data in these tissues would be a very helpful addition to the existing body of data. Unfortunately, these data are not available mostly for two reasons. Our experience with HiC (and similarly 4C-seq) is that generating data from tissues is much more complicated than using cell lines, especially when *very* large pools of tissue samples need to be collected, frozen, and later on combined (as would be the case here). Thus far, we have therefore been unable to generate HiC data from embryonic tissue samples. Further illustrating this, the fact that all published HiC data sets known to us (Liebermann-Aiden 2009, Dixon 2012, Jin 2013, Nagano 2013, Naumova, 2013, Sofueva 2013, Zuin 2013) are from cultured or circulating cells. The only exception is a mouse cortex HiC dataset (Dixon 2012) that is of much poorer quality due to low unique read numbers.

*5) The section about memorizing chromatin configurations is hard to follow, as it requires visual comparison of 4C data tracks in*
Figure 4
*with those in*
Figure 3*. Can the authors include a supplemental figure that displays these different profiles side-by-side (e.g., the E9.5 Tail bud tracks vs. the E10.5 Lumbo-sacral trum track and include a ratio plot)? The authors also need a more quantitative approach to compare such profiles, as the current comparison is based solely on visual inspections, and important differences between 4C profiles are rather subtle*.

We agree that the comparison is not very easy. This paper presents comparisons between several parameters (first between genes at X time points in Y tissues), then between the ‘same’ tissue at various time points. This is of course intrinsically difficult as we cannot guarantee that the cells indicated at T1 will be indeed those present in the tissue at T2. The major problem there is this that of the embryological lineage, which we approximate (there is not other way to do this more precisely). In this context, to have a precise comparison of the 4C profiles would not really help the case as this remains an approximation. Also, we did not want to display several times the same profile, even if it was to address different questions. We have tried to improve the text of this section too, to make it clearer for the reader.

*6) There are similarities of the model proposed by the authors in*
Figure 6
*and those with the work from Wendy Bickmore's laboratory where they have shown that the relocation of Hox clusters away from the*
*chromosome territory (CT) occurs during their activation […]*

We now refer to this model. However, the Bickmore laboratory has shown that the Hox clusters loop out the chromosome territory when activated, yet *not* following a sequential activation. The referees say: *‘’In the latter, sequential activation of the complex would involve its translocation from a region of repression (chromatin) to activation where there would be greater access to upstream regulatory factors’’.* We do not remember that this nice model involved a ‘sequential activation’. The cluster was either outside or inside, but this was not linked to progressive activation.

Also, the Bickmore lab showed that upon activation, the *HoxD* cluster loops out of its CT only along the developing trunk, but not in the early limb bud (Morey et al., Development 2007). We therefore consider it unlikely that looping from the CT is equivalent to TAD switching, which once again seems to occur only in limbs thus far.

*[…] or similarities to Spyros Papageorgiou's “cluster translocation model” hypothesis. In the former, the Hox complex is gradually excluded from CTs, which could correspond to the flanking TADs. In the latter, sequential activation of the complex would involve its translocation from a region of repression (chromatin) to activation where there would be greater access to upstream regulatory factors. Although, Spyros envokes morphogenic and electromotive (i.e., Coulomb force) components, nowhere do the authors mention these works in the content of their discussion or model*.

We have added references to these pieces of work now, yet we do not see anything in our paper, which can be ‘explained’ or ‘enlightened’ by the latter model. We nevertheless mention the electromotive possibility now.

Minor points:

*1) The statement that ES cells have a less compact Hox locus conformation are difficult to support by just 4C data. FISH might address this if it is available. The authors may wish to soften their conclusion otherwise*.

We did soften our wording and wherever the word ‘compact’ or ‘compaction’ was used, we now use the words ‘defined’, ‘discrete’ or ‘structuring’ instead:. We have also changed the text here and there to make it less explicit. The use of FISH to illustrate this point is certainly not appropriate, at least considering data available at this locus (Bickmore laboratory and our laboratory, unpublished). The level of resolution (the cluster is ca. 100 kb large) and the fact that in both cases some ‘compaction’ is observed makes it unlikely to give a decisive answer.

*2) The explanation of the apparent paradox between Hi-C and 4C data discussed at the end of the paper is reasonable, but a more likely explanation is that Hi-C simply has a lower resolution (20Kb at best). The authors suggest an apparent paradox when comparing Hi-C data and 4C data: the Hi-C data suggest a TAD boundary around d12 in ES cells, whereas d13 interacts with the rest of the Hox locus as observed by 4C. Their interpretation could be true, but in my view a more simple explanation is that the resolution of Hi-C (20 Kb at best) may be too low, and in the Dixon et al. data the true boundary can easily be 20Kb to the right or the left. I think the argument that Hi-C detects a boundary, while 4C does not, because Hi-C represents the average of two distinct conformations is not valid: 4C should be equally detecting this same average*.

We agree with the reviewers that the current resolution of HiC is not sufficient to precisely define the TAD border, but the paradox that we refer to is of a different nature: the four sampled *Hoxd* genes in ES cells are located in the same local 3D compartment, yet their ratio between centromeric *versus* telomeric contacts is highly graded (Figure 5, *Hoxd13* ∼70% of total TAD contacts in the centromeric TAD and *Hoxd4* ∼ 20% of total TAD contacts in the centromeric TAD – once more TADs as defined by [11]). Because the genes are in the same compartment (our 4C data), one would expect that the genes would act as a single interaction unit. Our preferred solution to this paradox remains therefore the solution we discuss in Figure 6. We have tried to clarify this paradox further in the text and figure legend.

This piece of discussion does not change the meaning of our results and was not strictly necessary for the readers. However, the fact is that a ‘boundary’ is seen in the ES cells HiC datasets, which we do not really observe when using 4C, unlike many other instances where our 4C data perfectly matched the TADs boundaries (Montavon et al., *Cell* 2011; Andrey et al., *Science* 2013). We think this is worth discussing, even though we do not have a clear explanation for it.

*3) In ES cells low basal expression of Hox genes is observed, and the locus appears to be in a single domain. The results certainly indicate that any local domain feature may be weaker in these cells as compared to forebrain. However, for the authors to conclude that this reflects a less compact organization they need FISH data*.

As mentioned under minor point 1, we have re-phrased all sentences including the word ‘compacted’. In our opinion, obtaining meaningful FISH data will be problematic, as current FISH data measured compaction between fully inactive *versus* partially activated clusters (Chambeyron et al, *Genes Dev* 2004 and Morey et al, *Development* 2007), which is equivalent to our single *versus* bimodal organizations. To compare two distinct states of ‘compaction’ is likely out of the current resolution of this methodology.

*Further, can the authors rule out that the increased outside-the-domain interactions in ES cells are not due to increased*
*random ligation in ES cells?*

This issue of increased random ligation as mentioned by the reviewers is indeed a valid point. We have added a new figure (Figure 1—figure supplement 6) where we compare the amount of signal in the surrounding TADs *versus* the rest of chromosome 2 or 11 (with the *HoxD/HoxB* clusters themselves excluded). As may be appreciated, the signal outside the surrounding TADs is in fact higher in terminally repressed forebrain cells. The increased interactions in ES cells are therefore limited only to the surrounding TADs. This excludes the possibility that the effect is due to random ligation. The consequences of these data are now discussed in the text.

*4) The authors find that genes switch autonomously between compartments. But it seems that d1-d8 act as a single domain, then D9-10-11 get included one by one, and d13 is always different (consistent with it being in a different domain based on Hi-C data). Perhaps the authors can either clarify this, or discuss this in the paper: do they think each gene is*
*acting independently, or are there three groups (D1-D8; D9-11, D13)?*

This is a very interesting issue indeed, which was briefly alluded to in the paper and which we tried to make clearer now. [50] reported that in the CNS, activation seems to work as a ‘two blocks’ strategy (gene 1 to 8 and genes 9 to 13 – the latter being part of the *AbdB* sub-group). This was one of the reasons why we carried out these experiments, to see if the chromatin ‘micro-domains’ were only two or alternatively, if they were progressively installed during trunk extension. We reached the conclusion that the latter situation is observed. Yet of course this is seen *only* in the d9 to d13 part of the cluster. To verify this on the anterior part would require thousands of dissections, *a fortiori* including cell sheets, which are not as well defined (homogenous) as in later embryos. For the moment, we think this experiment is not feasible. The referees are right in mentioning that our model may *not* apply to the anterior part (*d1* to *d8*) where sequential activation may *not* be paralleled by progressive chromatin transition. The text has been clarified accordingly.

*5) The data in*
Figure 5
*is puzzling: Hox d11 is switching domains within the Hox cluster from E8.5 to E10.5, yet it does have the same bias to interact with the telomeric desert as the d4 gene. Does this mean that d4 and d11 are really in the same larger topological domain at all time points, but that there is a change in sub-domain organizations? Cell type specific re-organization of the internal organization of topological domains has been proposed before (e.g., Phillips-Cremins et al. Cell 2013)*.

See under minor point 2. This is in fact the paradox we discuss in Figure 5. This is exactly the point; when negative, *Hoxd11* contacts the telomeric domain in long range, even though it does contact *Hoxd13*, which itself contacts the centromeric domain in long range. Considering that we have HiC data only for ES cells, we prefer to restrict the comparison to HiC *versus* 4C data in such ES cells. The only HiC data set from tissue (cortex HiC data set from [11]) does not really call a border inside the *HoxD* cluster, but rather tends to fuse the centromeric and telomeric ES-cell TADs into a large single TAD centered around the *HoxD* cluster.

*6) All results combined, one can imagine a constant global domain organization with temporal changes in sub-domains. It would be very informative to have high resolution (i.e., deeply sequenced) Hi-C maps for E8.5 PSMs, E10.5 Tail buds, and limb tissues. This would be a more useful Hi-C data set than the ES cells data from Dixon et al. I do realize this may be difficult given the amount of material that this may require, but it would be a very interesting dataset as it may reveal the stage-specific sub-domains as well as invariant larger domains in a single dataset, with potential changes in the larger domains in limb*.

As discussed under major point 4, we agree with the reviewers that this data would be very insightful. Yet we (and most likely others) have not been able to generate such data from embryonic tissues. Also, HiC data from E8.5 mouse embryos is not yet a reasonable target. Our estimate indeed would be that between 2’000 and 4’000 E8.5 PSM dissections would be required to generate a single HiC library.

*7) In the legend of*
Figure 6*, the authors indicate that the 'apparent boundary' near the Hoxd11 gene is indicated by grey lines but do they not mean the dashed blue line passing from*
Figure 6
*into 6B? Or are they referring to the grey shaded triangle projecting from the middle of Hoxd11 onto the cartoon of the HoxD*
*complex below it?*

The mentioned grey lines are the two diagonal lines in the triangle representing the HiC data. These lines originate from the same point as where the dashed line (which is grey as well) originates. To clarify this issue, we have rewritten the legend, which now reads: “(A) Schematic organization of topological domains in ES cells [from (11)] matching the centromeric and telomeric gene deserts, with an apparent boundary assigned near the *Hoxd11* gene (grey diagonal lines). All *Hoxd* genes in ES cells have considerable interactions on either side of the cluster, suggesting that this border is more diffuse and hence the entire *HoxD* cluster can be integrated in either TAD (diagonal black lines).”

*8) In regards to the replicates presented in supplemental data to*
Figure 1*, where the contacts within and outside the HoxD complex are represented as a graphical ratio between ES cells and forebrain cells: visually this conveys the difference in the 3D organization of the inactive Hox complex between the ES cells and forebrain cells, however between replicates there is surprising variation in the ratio. Would we not expect less variation in the produced ratios between replicates which are produced from pooled samples of ES cells and forebrain cells? In the absence of the associated chromatin marks, this makes it hard to argue that in ES cells the HoxD cluster is less defined than in forebrain cells. Furthermore, one would worry that it would be easy to pick data from one replicate over another to fit the hypothesis regarding the remodeling of the HoxD architecture during sequential gene activation*.

It is correct that quite some variation can be observed between the replicates. Partially, this is due to the commonly observed variation at the individual fragment level in 4C experiments, combined with the log2 scale that is used to depict ratios. Relatively small changes therefore already can appear quite prominent in the graph. However, we initially shared the concerns of the reviewers and thus included two more safeguards in the analysis:

1) To overcome the effect of signal variation at the single fragment level, we rather decided to assess the change within the *Hox* complexes *versus* outside the *Hox* complexes. Our rationale here was that if the changes were purely experimental, no regional difference would be expected. To further exclude variation at the single fragment level, we decided not to look at the value of the ratio, but rather just at the sign (positive or negative). We realize that in our previous manuscript, the figures with these data were too small. In the current Figure 1—figure supplement 5, the data can be now better appreciated. For all those viewpoints where we compare in-*versus*-out for ES cells versus brain cells, we find a significant regional difference. For all those viewpoints, we can therefore conclude that ES cells have a significantly different distribution *versus* forebrain cells, when assessing their signal within their *Hox* cluster *versus* outside their *Hox* cluster.

2) To further rule out experimental variation, the replicate samples (replicate 2) were generated as true technical replicates and processed simultaneously with each step done using the same batch or reagents. The resulting data set was unfortunately not of the same high quality as the data from replicates 1 and this is primarily reflected by a lower number of genomic fragments outside of the *Hox* clusters with signal. Overall, these data sets still displayed the same pattern of interactions (see Table 4), but with less significant p-values due to the fewer useful fragments in the analysis. This is in contrast to the comparison between replicates, which is generally not significantly different (or, in the case of the *Hoxb13* gene ES rep 1 *versus* ES rep 2: very marginally). Within the data set, we do realize there is one large outlier: the comparison between the *Hoxd13* replicate 1 *versus* replicate 2. Within the overall replicate 2 experiment, this particular viewpoint was most problematic. To keep consistency within the experiment, and not hand pick which data to include, we decided to maintain it. Importantly, despite it being significantly different, the differences between ES cells and forebrain cells are always much more significant.Author response table 1.View pointES rep 1 vs FB rep 1ES rep 2 vs FB rep 1ES rep 1 vs FB rep 2ES all vs FB allES rep 1 vs ES rep 2FB rep 1 vs FB rep 2*Hoxd13*9.92E - 211.70E - 117.63E - 082.21E - 340.591.09E - 04*Hoxd9*3.02E - 020.360.912.61E - 030.900.27*Hoxd4*1.90E - 090.0150.0141.36E - 110.780.11*Hoxb13*1.37E - 047.65E - 030.0104.42E - 140.0370.51*Hoxb9*2.75E - 070.0190.0122.21E - 090.200.25p-values of difference in distribution between replicate samples. ES: ES cells, FB: forebrain.

To further rule out that the difference in discretion is an experimental effect, we have analyzed the replicate data in the same way as done for Figure 1—figure supplement 6 to Figure 2 (discussed in Minor point 3 of this rebuttal). This figure (Figure 8) hints that an increase of non-specific interactions may be the cause of the poorer quality of the replicate 2 data set (for all rep 2 samples the ‘distal chromosome’ category is increased). Importantly though, for all viewpoints the ES cell samples have increased interactions in the TADs in comparison to the forebrain data sets. These data therefore further support our finding that the compartments on the *Hox* clusters are less discrete in ES cells than in terminally repressed forebrain cells.Author response image 2.